# Commensal *Escherichia coli* Strains of Bovine Origin Competitively Mitigated *Escherichia coli* O157:H7 in a Gnotobiotic Murine Intestinal Colonization Model with or without Physiological Stress

**DOI:** 10.3390/ani13162577

**Published:** 2023-08-10

**Authors:** Maximo E. Lange, Sandra T. Clarke, Valerie F. Boras, Catherine L. J. Brown, Guangzhi Zhang, Chad R. Laing, Richard R. E. Uwiera, Tony Montina, Martin L. Kalmokoff, Eduardo N. Taboada, Victor P. J. Gannon, Gerlinde A. S. Metz, John S. Church, G. Douglas Inglis

**Affiliations:** 1Lethbridge Research and Development Centre, Agriculture and Agri-Food Canada, Lethbridge, AB T1J 4B1, Canada; maximo.lange@agr.gc.ca (M.E.L.); sandra.clarke2@agr.gc.ca (S.T.C.); kbrownbiotech@gmail.com (C.L.J.B.); 2Department of Agricultural, Food and Nutritional Science, University of Alberta, Edmonton, AB T6G 2R3, Canada; ruwiera@ualberta.ca; 3Chinook Regional Hospital, Alberta Health Services, Lethbridge, AB T1J 1W5, Canada; vfboras@live.ca; 4National Microbiology Laboratory, Public Health Agency of Canada, Winnipeg, MB R3E 3R2, Canada; guangzhi.zhang@phac-aspc.gc.ca (G.Z.); eduardo.taboada@phac-aspc.gc.ca (E.N.T.); 5National Centre for Animal Diseases, Canadian Food Inspection Agency, Lethbridge, AB T1J 3Z4, Canada; chad.laing@inspection.gc.ca; 6Department of Chemistry and Biochemistry, University of Lethbridge, Lethbridge, AB T1K 3M4, Canada; tony.montina@uleth.ca; 7Kentville Research and Development Centre, Agriculture and Agri-Food Canada, Kentville, NS B4N 1J5, Canada; mkalmoko@outlook.com; 8National Microbiology Laboratory, Public Health Agency of Canada, Lethbridge, AB T1J 3Z4, Canada; gannvp@yahoo.ca; 9Canadian Centre for Behavioural Neuroscience, Department of Neuroscience, University of Lethbridge, Lethbridge, AB T1K 3M4, Canada; gerlinde.metz@uleth.ca; 10Natural Resource Science, Thompson Rivers University, Kamloops, BC V2C 0C8, Canada; jchurch@tru.ca

**Keywords:** mouse, germ-free, competitive exclusion, calf model, gut

## Abstract

**Simple Summary:**

Prescribed examination of enterohemorrhagic *Escherichia coli* (EHEC) O157:H7 microbiota–host interactions in the gastrointestinal tract of cattle is technically difficult due in part to the high cost of conducting research with cattle, the genetic heterogeneity among animals, the logistic challenges of obtaining prescribed samples, and the variability in the structure of the enteric microbiota among individuals. Thus, our overarching goal was to develop a prescribed enteric colonization model utilizing germ-free mice inoculated with individual bovine EHEC O157:H7 strains representing the primary genetic lineages of the pathogen. Moreover, we utilized the colonization model with or without stress induced via the administration of corticosterone to examine the ability of commensal *E. coli* strains to outcompete EHEC O157:H7 in vivo. A bovine strain of EHEC O157:H7 that incited reduced pathologic changes was identified, and the administration of 18 commensal *E. coli* strains isolated from cattle effectively reduced densities of the pathogen, and ameliorated histopathologic changes and markers of inflammation. Although stress has been identified as a factor affecting colonization success, we observed that physiological stress did not benefit enteric colonization by EHEC O157:H7. Despite its limitations, the defined microbiota murine enteric colonization model developed may prove useful for identifying mechanisms and mitigation strategies for subsequent validation in cattle.

**Abstract:**

Cattle are a primary reservoir of enterohemorrhagic *Escherichia coli* (EHEC) O157:H7. Currently, there are no effective methods of eliminating this important zoonotic pathogen from cattle, and colonization resistance in relation to EHEC O157:H7 in cattle is poorly understood. We developed a gnotobiotic EHEC O157:H7 murine model to examine aspects of the cattle pathogen–microbiota interaction, and to investigate competitive suppression of EHEC O157:H7 by 18 phylogenetically distinct commensal *E. coli* strains of bovine origin. As stress has been suggested to influence enteric colonization by EHEC O157:H7 in cattle, corticosterone administration (±) to incite a physiological stress response was included as an experimental variable. Colonization of the intestinal tract (IT) of mice by the bovine EHEC O157:H7 strain, FRIK-2001, mimicked characteristics of bovine IT colonization. In this regard, FRIK-2001 successfully colonized the IT and temporally incited minimal impacts on the host relative to other EHEC O157:H7 strains, including on the renal metabolome. The presence of the commensal *E. coli* strains decreased EHEC O157:H7 densities in the cecum, proximal colon, and distal colon. Moreover, histopathologic changes and inflammation markers were reduced in the distal colon of mice inoculated with commensal *E. coli* strains (both propagated separately and communally). Although stress induction affected the behavior of mice, it did not influence EHEC O157:H7 densities or disease. These findings support the use of a gnotobiotic murine model of enteric bovine EHEC O157:H7 colonization to better understand pathogen–host–microbiota interactions toward the development of effective on-farm mitigations for EHEC O157:H7 in cattle, including the identification of bacteria capable of competitively colonizing the IT.

## 1. Introduction

*Escherichia coli* O157:H7 is a foodborne pathogen of human beings for which cattle are considered to be the primary reservoir [1]. Human beings infected with enterohemorraghic *E. coli* (EHEC) O157:H7 suffer non-bloody diarrhea, bloody diarrhea, and potentially develop hemolytic uremic syndrome (HUS) [2]. Shiga toxin (Stx)-producing *E. coli* are estimated to cause >2.8 million acute illnesses, 3,890 cases of HUS, 270 cases of permanent end-stage renal disease, and 230 deaths per year globally [3]. It is not entirely clear at present how EHEC O157:H7 successfully colonizes and survives in the intestine of cattle without causing clinical illness. Furthermore, the immune response mounted by bovine hosts, and colonization locations and shedding patterns of the bacterium are poorly understood [4,5]. There are currently no definitive prevention strategies for eliminating EHEC O157:H7 in its natural bovine reservoir.

Cattle are exposed to multiple stressors during production including weaning, vaccination, dietary changes, transportation, and confinement, among others. Different handling and management procedures have been shown to increase levels of the glucocorticoid stress hormone, cortisol, in cattle [6,7,8]. Stressful events have been implicated as a potential risk factor responsible for enhancing the shedding of EHEC O157:H7 in feces, particularly for calves early in the feeding period [9,10,11]. It is well established that stress can have an array of effects on the immune system [12,13], and this may potentially benefit EHEC O157:H7 colonization and persistence in cattle [9]. Furthermore, stress hormones such as catecholamines have been demonstrated to enhance the expression of virulence genes in EHEC O157:H7 that directly benefit its colonization [14,15]. At present, the degree and mechanisms by which physiological stress can potentially influence the intestinal colonization by EHEC O157:H7 are poorly understood and require further investigation.

Elucidating the cardinal factors controlling intestinal colonization in cattle is challenging. In this regard, the complexity and variability of the intestinal microbiota among animals makes it difficult to mechanistically study EHEC O157:H7 colonization, including host–bacteria interactions, and competition amongst bacteria. Additionally, husbandry practices and the costs involved in utilizing cattle in experiments can be a limiting factor for many researchers. Consequently, specific-pathogen-free and streptomycin dysbiosis murine models have been used to investigate EHEC O157:H7 colonization, but the background noise generated by the intestinal microbiota in these models limits the ability of researchers to make definitive conclusions on interactions between bacteria and the host [16,17,18,19]. Since a highly representative cattle model to elucidate key aspects of the host–pathogen–microbiota interaction does not yet exist, we utilized mice as an intestinal colonization model for the research presented herein [20]. More specifically, we used a gnotobiotic (GB) C57BL/6 murine model to investigate the impacts of physiological stress on a population of commensal *E. coli* strains from cattle on host and microbial responses, including competitive colonization by bovine EHEC O157:H7. Importantly, GB animals possess a microbiota in which all microorganisms associated with the animal are known, whereas germ-free (GF) animals are devoid of microorganisms. Therefore, GF mice become GB when they are inoculated with defined bacterial taxa. From a human-health perspective, the interaction between EHEC O157:H7, the host, and other bacteria have been previously studied in GB mice and other murine models [21,22,23,24]. However, to our knowledge, these interactions from a cattle perspective, including the effects of stress, have not been previously examined. Additionally, the immunological responses and colonization impacts of a bovine EHEC O157:H7 in a murine model of stress have not been previously studied.

Five different EHEC O157:H7 strains representing different lineages were initially evaluated in a GB murine model to assess intestinal colonization patterns, pathogenicity, and the virulence of the strains. Secondly, we examined the degree to which an EHEC O157:H7 strain from cattle could compete with 18 bovine commensal *E. coli* strains in GB mice administered exogenous corticosterone as an incitant of physiological stress. Commensal strains grown separately and communally were contrasted. We hypothesized that stress predisposes the host by altering the immune response and directly influencing bacterial virulence factors, thereby providing EHEC O157:H7 with a competitive advantage over commensal strains of *E. coli* that occupy the same enteric niche as EHEC O157:H7 (i.e., competitive colonization).

## 2. Materials and Methods

### 2.1. Mice

All mice were produced from a breeding colony of GF C57BL/6 mice maintained at Agriculture and Agri-Food Canada (AAFC) Lethbridge Research and Development Centre (LeRDC), which were generated from adult mice obtained from the National Gnotobiotic Rodent Resource Center (NGRRC) at the University of North Carolina. GF C57BL/6 mice were reared and maintained in a Flexible Film GF isolator (Class Biologically Clean Ltd., Madison, WI, USA), and were transferred to and housed in Tecniplast^®^ Green Line GM500 Sealsafe Plus Individually Ventilated Cages (IVC) (Tecniplast, Toronto, ON, Canada) situated in a single-sided Sealsafe Plus rack (Tecniplast) attached to a Smart Flow (Tecniplast) high efficiency particulate air (HEPA) filter unit operated in containment mode (i.e., negative air pressure circulation) accordingly to the manufacturer’s recommendations [25]. Sterilized enrichment items were placed in each cage. The IVCs were situated within a vivarium in animal rooms operated in containment mode (inward directional airflow), with unfiltered ambient air entering and exiting the room. Methods for housing and handling of GF mice in IVCs are described by Lange et al. [25]. Mice were maintained on a 12 h light/dark cycle and allowed to drink and eat ad libitum.

To confirm the GF status of mice, feces from non-inoculated GF mice were weighed, suspended in 1 mL sterile of phosphate-buffered saline (PBS, pH 7.2) and vortexed (high setting) for 20 s. Aliquots of the suspension (25 µL) were spread in quadruplicate onto Columbia agar (Difco; Becton Dickinson Canada Inc., Mississauga, ON, Canada) containing 5% sheep blood. Half of the cultures were incubated in an anoxic atmosphere (9–13% CO_2_, with less than 0.1% O_2_) at 37 °C in 2.5 L anaerobic jars (Oxoid™ AnaeroJar™ 2.5L, Thermo Scientific™, Ottawa, ON, Canada), and the other cultures were incubated in an aerobic atmosphere at 37 °C. After 7 days, cultures were examined for microbial growth.

### 2.2. Escherichia coli Strains

The following five EHEC O157:H7 strains representing different phylogenetic groups were evaluated in the study: (1) EDL933 (strain responsible for the first outbreak in people); (2) FRIK-2001 (bovine isolate); (3) TW14359 (hyper-virulent strain isolated from a human being); (4) ECI-1375 (bovine isolate); and (5) ECI-1911 (bovine isolate).

Eighteen commensal *E. coli* isolates recovered from beef cattle in Alberta and Nova Scotia were evaluated. These isolates were phenotypically determined to not produce Stx, and they were selected based on their ability to competitively exclude EHEC O157:H7 in a chemostat (unpublished data). To determine that the commensal *E. coli* isolates represented unique subtypes, they were initially genotyped by pulsed-field gel electrophoresis (PFGE) using the protocol specified by the Centers for Disease Control and Prevention (CDC) [26]. Briefly, the enzyme XbaI (New England Biolabs Canada, Whitby, ON, Canada) was used for restriction endonuclease digestion, and electrophoresis was executed with a CHEF-DR^®^ III PFGE System (Bio-Rad Laboratories Inc., Hercules, CA, USA) using a 1% agarose gel. The electrophoresis conditions were as follows: initial switch time 2.2 s, final switch time 54.2 s, voltage 6 V, included angle 120°, flow rate 1 L/min, and run time 19 h at 14 °C. Gel images were captured utilizing an AlphaImager 2200 (Alpha Innotech, San Leandro, CA, USA) and analyzed with BioNumerics 6.6 (Applied Maths, Sint-Martens-Latem, Belgium). A reference *E. coli* commensal strain (LCMB-18-J) was used as a standard.

### 2.3. Design and Validation of Primers to Detect and Quantify EHEC O157:H7 Strains

Primers were designed from whole-genome sequencing (WGS) data for the five EHEC O157:H7 strains [27]. Regions unique to the genome of each of the five strains were identified utilizing Panseq (using default settings). Specific primers were designed utilizing Geneious 5.3.6 (Biomatters, Boston, MA, USA) (Appendix A) targeting the putative unique sequences of each strain’s genome (i.e., relative to each other). The specificity of all designed primers was determined using endpoint PCR using genomic DNA extracted from cells of each strain in the late log stage of growth using a DNeasy^®^ Blood and Tissue Kit (Qiagen Inc., Toronto, ON, Canada) according to the manufacturer’s specifications. Reaction mixtures consisted of a total volume of 20 µL containing the following: 2 µL of genomic DNA, 2 µL of reaction buffer (Qiagen Inc.), 0.4 µL of deoxynucleoside triphosphates (0.2 mM), 0.4 µL of MgCl_2_ (2 mM), 1 µL of each primer (0.5 µM; Integrated DNA Technologies, Coralville, IA, USA), 0.1 µL of HotStar Taq polymerase (Qiagen Inc.), and 13.1 µL of nuclease-free water (Qiagen Inc.). The PCR cycle conditions were as follows: one activation cycle at 95 °C for 5 min; 35 cycles at 94 °C for 15 s, 62 °C for 30 s, and 72 °C for 30 s; and a final extension at 72 °C for 5 min. The primers were also designed for quantitative PCR (qPCR). The following reagents were used: 2.0 μL of DNA, 10 μL of 2× QuantiTect^®^ SYBR^®^ Green Master Mix (Qiagen Inc.), 1.0 μL of the forward and reverse primer (0.5 μM; Integrated DNA Technologies), and 6.0 μL of nuclease-free water (Qiagen Inc.). The qPCR conditions were as follows: one activation cycle at 95 °C for 15 min; 40 cycles at 94 °C for 15 s, and 68 °C for 30 s; and melt-curve analysis. A Mx3005p Real Time PCR instrument (Agilent Technologies Canada Inc., Mississauga, ON, Canada) was used. To calculate a density value, threshold cycles (Ct values) for each sample were compared to a standard curve generated from of known quantities of DNA extracted from each EHEC O157:H7 strain. One primer set that was determined to be specific for each EHEC O157:H7 strain was selected. The specificity of the primers was further evaluated against DNA of the 18 commensal *E. coli* strains isolated from beef cattle, and none of the EHEC O157:H7 primers produced an amplicon in endpoint or qPCR.

### 2.4. Design and Validation of Primers to Detect and Quantify Commensal Escherichia coli Strains

Strain-specific sequences for the 18 commensal *E. coli* strains were obtained using whole-genome sequencing and whole-genome multi-locus sequence typing (wgMLST) analysis. WGS of the commensal *E. coli* strains was conducted using an Illumina NextSeq500 platform (Illumina, San Diego, CA, USA) with Nextera libraries through a next-generation sequencing core facility (National Microbiology Laboratories, Winnipeg, MB, Canada). Genome assembly and wgMLST analysis were performed using the WGS tool client plug-in in BioNumerics^TM^ version 7.6.3 (Applied Maths) with default parameters, using a scheme containing 15,136 loci. wgMLST allele detection was performed using either assembly-free allele calling with a kmer size of 35 and assembly-based calling in which de novo assembly was first performed with SPAdes version 3.7.1, followed by similarity searching for reference alleles using the Basic Local Alignment Search Tool (BLAST). Of the unique alleles for each strain obtained by wgMLST analysis, three were selected for qPCR primer design. The nucleotide sequences of the selected alleles were submitted to the PrimerQuest Tool, IDT (https://www.idtdna.com/Primerquest (accessed on 6 August 2023)), setting the design parameters as qPCR Intercalating Dyes (Primers only). For each allele, five primer pairs were obtained. To verify the stain specificity of the primer pairs, the candidate primers were searched for similarity against a database of draft genomes of the 18 commensal *E. coli* strains with BLAST using Geneious Primer (Biomatters; https://www.geneious.com (accessed on 6 August 2023)). Primers that showed no off-target annealing among the bacterial sequences were retained for experimental verification. To complete experimental primer testing, qPCR was completed in triplicates as follows: 1.0 μL of DNA, 5.0 μL of 2× QuantiTect^®^ SYBR^®^ Green Master Mix (Qiagen Inc.), 0.5 μL of the forward and reverse primer (0.5 μM; Integrated DNA Technologies), and 3.0 μL of nuclease-free water (Qiagen Inc.). The PCR conditions were as follows: one activation cycle at 95 °C for 15 min; 40 cycles at 94 °C for 15 s, at the annealing temperature specific to each primer (Appendix A) for 30 s, and 72 °C for 30 s; and melt-curve analysis. A Quantstudio Real Time PCR instrument (Thermo Fisher Scientific, Ottawa, ON, Canada) was used. The specificity of the primer for each strain was evaluated against the other commensal *E. coli* strains, as well as the EHEC O157:H7 strains. No non-target amplicons were produced.

### 2.5. Whole Genome Phylogenetic Analysis of Commensal Escherichia coli Strains

Core-genome (cg)MLST and whole-genome single-nucleotide polymorphism (wgSNP) clustering were conducted in BioNumerics version 7.6.3 (Applied Maths) using the categorical-difference coefficient with a scaling factor of 100. Dendrograms were built using the UPGMA algorithm. For whole-genome variant calling, the *E. coli* reads were reference mapped to Sakai (GenBank Accession: BA000007.2) using the WGS plug-in. The pairwise wgSNP similarity matrix was then generated and used to produce the wgSNP dendrogram. The cgMLST (EnteroBase schema) consists of 2,513 core loci; the pairwise similarity matrix based on these loci was used for tree production.

### 2.6. Commensal Escherichia coli Strain In Silico Phenomic Predictions

Genome assemblies were subjected to analysis using a suite of tools that included the following: “mlst” for MLST prediction (https://github.com/tseemann/mlst (accessed on 6 August 2023)); ECTyper for serotype prediction [28]; the EzClermont online phylotyping prediction tool [29]; Staramr for antimicrobial resistance prediction [30]; and the VirulenceFinder 2.0 online tool for virulence factor prediction [31,32]. Analyses made use of the PubMLST website (https://pubmlst.org/ (accessed on 6 August 2023)) developed by Jolley and Maiden [33].

### 2.7. Colonization and Virulence of EHEC O157:H7 Strains in Mice

The colonization characteristics and virulence of the five different EHEC O157:H7 strains were determined in GB mice. Male GF mice (4-to-6-week-old) were transferred from the GF isolators into IVCs and acclimated for a period of 1 week at which point they were inoculated with EHEC O157:H7 strains. The experiment was designed as a two (time point) by six (*E. coli* treatment) factorial with three replicates conducted on separate occasions (twelve mice per replicate). Mice were administered the following EHEC O157:H7 strains/treatments: (1) EDL933; (2) FRIK-2001; (3) TW14359; (4) ECI-1375; (5) ECI-1911; or (6) PBS alone (i.e., no bacteria control treatment). Mice were orally inoculated with bacteria or PBS alone on day 0, and were humanely euthanized 5 and 8 days post-inoculation (p.i.) (see Section 2.10 for inoculum-administration details).

### 2.8. Competitive Colonization by EHEC O157:H7 in Mice under Physiological Stress

The ability of the EHEC O157:H7 strain, FRIK-2001, to competitively colonize mice under conditions of physiological stress was determined. The experiment was conducted as a two (±corticosterone) by two (±EHEC O157:H7) by three (commensal *E. coli* treatment) factorial (Appendix A). Three replicates were conducted on separate occasions (twelve mice per replicate). Male GF mice (4-to-8-weeks-of-age) were transferred from GF isolators into IVCs and permitted to acclimate for a period of 1 week before commencement of the experiment. For the corticosterone treatment, mice were arbitrarily assigned to one of two groups; one group was administered corticosterone in water to induce physiological stress, and the second group was provided drinking water free of the glucocorticoid. It is noteworthy that corticosterone is a key hormone released in stress situations in mice and has been used to induce stress as well as a metric of stress [13,34,35,36,37]. Corticosterone treatment commenced on day 0 and continued throughout the 9-day experimental period (see Section 2.11 for corticosterone administration details). On day 6, mice were inoculated with EHEC O157:H7 (±) and/or commensal *E. coli* strains (±). The commensal *E. coli* treatments were as follows: (1) mice inoculated with 18 commensal *E. coli* strains grown communally (CC); (2) mice inoculated with the same 18 commensal *E. coli* strains grown separately (CS); and (3) mice not inoculated with commensal *E. coli* (Control treatment) (see Section 2.10 for inoculum-administration details). The rationale for comparing *E. coli* isolates grown communally vs. separately was based on a previous study in which selected mixtures of bacteria grown communally were more effective than the same bacteria propagated separately for competitively excluding *Salmonella enterica* serovar Typhimurium from the intestinal tract of chickens [38]. Behavioral assessments followed by euthanization of mice were conducted on day 9 (i.e., 3 days p.i.); the 3-day p.i. endpoint was chosen based on the findings from experiment one in which mice showed a limited response to FRIK-2001 before 5 days p.i. (i.e., mimicking cattle).

### 2.9. Propagation of EHEC O157:H7 and Commensal Escherichia coli Strains

All *E. coli* strains were grown aerobically in 20 mL of Luria-Bertani (LB; Sigma Aldrich, Oakville, ON, Canada) broth at 37 °C while shaking at 115 rpm to a mid-logarithmic phase of growth as determined by optical density at 600 nm. The cultures were centrifuged at 12,000× *g* for 5 min, the supernatant was removed, and bacterial cells were re-suspended in PBS to a final concentration of 1 × 10^6^ cells/mL. The density of cells was confirmed by diluting the cell suspension in a 10-fold series, spreading 100 µL onto LB agar in duplicate, and counting colonies at the dilution yielding 30 to 300 colonies.

Commensal *E. coli* were grown separately or communally. For commensal *E. coli* strains grown separately, each isolate was grown in 20 mL of LB broth, the medium was removed by centrifugation, and cells were re-suspended in PBS as above. The *E. coli* strains were combined by pooling 1 mL of each culture in PBS together to achieve a final concentration of 1 × 10^6^ CFU/mL. For commensal *E. coli* strains grown communally, one colony of each strain was placed into a common tube with 35 mL of LB broth. After 5 h of growth at 37 °C, 5 mL of culture was removed, centrifuged, and the supernatant removed. The pellet was re-suspended in 45 mL of PBS to achieve a final concentration of 1 × 10^6^ CFU/mL, which was confirmed using the dilution spread-plate method. To determine the relative abundance of commensal strains grown communally in LB both (three cultures), PFGE was used. On the day of cell harvest for the inoculation of mice, the culture broth was diluted in a ten-fold series, and 100 µL of each dilution was spread onto LB agar. After 24 h growth at 37 °C, biomass from 100 arbitrarily selected colonies from each of the three cultures was collected (300 total), isolates were propagated in LB broth, and biomass was stored in LB with 30% glycerol at −80 °C. The *E. coli* isolates were grown from frozen stocks on Sorbitol MacConkey agar (Sigma Aldrich), and their fingerprints were obtained by PFGE and analyzed using BioNumerics version 6.6 (Applied Maths). Assignments to individual strains were accomplished using PFGE fingerprints obtained from pure cultures.

### 2.10. Inoculation of Mice with EHEC O157:H7 and Commensal Escherichia coli Strains

*Escherichia coli* were administered to mice in sterile raspberry Jell-O (Kraft-Heinz Canada, Don Mills, ON, Canada). A suspension of bacterial cells or PBS alone (100 µL) was uniformly mixed into 7 mL of sterile Jell-O placed in sterile 60-mm-diameter Petri dishes. In instances where EHEC O157:H7 and commensal *E. coli* strains were inoculated together, cell suspensions of both were mixed into the Jell-O at the same time. A single Petri dish was placed in the IVC containing a single mouse. All mice consumed the Jell-O within 1–2 h. This method of inoculation was selected to reduce handling and the risk of compromising the GB status of the mice.

### 2.11. Corticosterone Administration

To induce stress, mice were administered corticosterone (100 µg/mL; Sigma Aldrich, Oakville, ON, Canada) in sterile drinking water [36,39]. The corticosterone was dissolved in absolute ethanol before addition to the water placed in conventional murine water dispensers (1% final ethanol concentration *v*/*v*). Control treatment mice were administered water containing ethanol alone (1% *v*/*v*). Mice were allowed to drink ad libitum.

### 2.12. Health Assessments

Following the administration of corticosterone and inoculation with EHEC O157:H7, mice were scored each morning at the same time (i.e., 9:00 am) for general activity (0–4), hair coat appearance and grooming behavior (0–3), and vocalization (0–1) (Appendix A). The health assessment score informed decisions on whether to humanely euthanize mice (i.e., in advance of planned experimental endpoints as warranted).

### 2.13. Behavioral Assessments

For experiment two, enrichment items in IVCs were removed from the cages 2.5 days p.i., and mice were aseptically provided 3.0 g of a sterile cotton nestlet. The nestlet was supplied 1 h before the dark phase. Nest building quality was assessed the next morning following a 1 to 5 rating scale [40]. On day 3 p.i. (i.e., immediately prior to euthanization), an open field test was performed. A 30 × 30 cm Phenotyper cage (Noldus Information Technology Inc., Leesburg, VA, USA) was utilized to record exploratory behaviour. Each mouse was placed in the center of the cage, and the behaviour was recorded for 10 min. Personnel remained behind curtains during the video recording to minimize environmental distractions. All videos were analyzed with Ethovision XT10 (Noldus Information Technology Inc.), and measurements of center zone frequency, cumulative time in center, latency to first in center, total distance moved, and velocity of movement were quantified.

### 2.14. Sample Collection

At experimental endpoints, or as informed by the health assessment score (Appendix A), mice were anesthetized with isoflurane, and blood was collected by cardiac puncture. Blood for serum separation was collected into BD Microtainer^®^ SST tubes (BD, Franklin Lake, NJ, USA), and serum for quantification of corticosterone was stored at −80 °C until analyzed. Under anesthesia, mice were then humanely euthanized by cervical dislocation. A mid-line laparotomy was completed with sterile tools to exteriorize the intestine. Sections from the kidney, ileum, cecum, and proximal and distal colon were removed. Within ca. 5 min of death, samples for gene expression were placed in RNAprotect^®^ (Qiagen Inc.) and stored at −80 °C. Tissue samples for quantitative metabolomics were snap frozen in liquid nitrogen, and subsequently stored at −80 °C. Samples for histopathologic examination were placed in 10% neutral buffered formalin (Leica Microsystems, Concord, ON, Canada) at room temperature. In addition, intestinal biopsies (4-mm-diameter) were obtained from the cecum, proximal colon, and distal colon, and stored at −80 °C for enumeration of *E. coli* by qPCR. Subsamples of digesta (180–220 mg) from the ileum and cecum were also collected, and where possible, from the proximal colon and distal colon, and stored at −80 °C for enumeration of *E. coli* by qPCR and to quantify corticosterone.

### 2.15. Histopathologic Scoring

Sections of the cecum, proximal colon, and distal colon fixed in 10% neutral buffered formalin were dehydrated using a Leica tissue processor (Leica TP1020 Benchtop Tissue Processor, Leica Biosystems, Concord, ON, Canada). Following dehydration, tissues were embedded in paraffin using a Shandon Histocentre 3 Embedding Center (Thermo Fisher Scientific, Ottawa, ON, Canada), sectioned (≈5 µm) using a Finesse 325 Manual Rotary Microtome (Thermo Fisher Scientific), and the sections were placed on 25 × 75 × 1 mm Superfrost Plus Gold microscope slides (Fisher Scientific, Ottawa, ON, Canada) and deparaffinized. The slides were then stained with hematoxylin and eosin, and sections (mucosa and submucosa) were examined using a Zeiss Axioskop Plus microscope (Carl Zeiss Canada Ltd., North York, ON, Canada). Histopathologic changes were scored for cell-infiltrate severity (1–4) and extent (1–3), crypt elongation (1–5), epithelial injury (1–4), cryptitis (2–3), crypt abscesses (4–5), goblet-cell loss (1–4), granulation tissue (4–5), crypt loss (4–5), apoptosis (0–3), occluding thrombi (0–3), mucosal hemorrhage (0–3), irregular crypts (4–5), villar blunting (1–5), and ulceration (3–5), as described previously [41,42,43]. Total scores were calculated by combining the scores for all metrics (possible total maximum score of 62). Samples were scored by a board-certified pathologist (V.F.B.) who was blinded to treatment.

### 2.16. Detection and Quantification of EHEC O157:H7

The densities of EHEC O157:H7 associated with mucosa and within digesta were quantified by qPCR. Genomic DNA was extracted from 80–120 mg of digesta samples using the QIAamp^®^ Fast DNA stool Kit (Qiagen Inc.) according to the manufacturer’s recommendations. Genomic DNA from mucosal biopsies was extracted using the DNAeasy^®^ Blood and Tissue Kit (Qiagen Inc.). qPCR was conducted as described above.

### 2.17. Detection and Quantification of Commensal Escherichia coli

Genomic DNA was extracted from colonic digesta, and the presence and densities of each commensal *E. coli* strain were measured by qPCR as described above.

### 2.18. Quantification of Total Escherichia coli

Genomic DNA was extracted from colonic digesta, and densities of total *E. coli* were measured by qPCR. Genomic DNA was extracted from 80–120 mg of thawed digesta samples using the QIAamp^®^ Fast DNA Stool Mini Kit (Qiagen Inc.) according to the manufacturer’s recommendations. qPCR was completed with the universal bacterial primer set, HDA1/HDA2 (F: ACTCCTACGGGAGGCAGCAGT; R: GTATTACCGCGGCTGCTGGCAC) [44], and was completed in triplicate as follows: 1.0 μL of DNA, 5.0 μL of 2× QuantiTect^®^ SYBR^®^ Green Master Mix (Qiagen Inc.), 0.9 μL of forward primer, 0.6 μL of reverse primer (10 μM; Integrated DNA Technologies), and 2.5 μL of nuclease-free water (Qiagen Inc.). The qPCR conditions were as follows: one activation cycle at 95 °C for 15 min; 40 cycles at 94 °C for 15 s, 56 °C for 30 s, and 72 °C for 30 s; a final cycle at 95 °C for 1 min; 55 °C for 1 min; and melt-curve analysis. A Quantstudio Real Time PCR instrument (Thermo Fisher Scientific) was used.

### 2.19. Quantification of Inflammation Gene mRNA

To quantify the mRNA of targets of interest, RNA was extracted from ≈0.5 × 0.5 cm sections of distal colon using a RNeasy^®^ Mini Kit (Qiagen Inc.) with a DNase step added to eliminate residual genomic DNA. RNA quantity and quality was determined using a Bioanalyzer 2100 (Agilent Technologies Canada Inc.). RNA (1000 ng) was transcribed into cDNA using a QuantiTect^®^ Reverse Transcription Kit (Qiagen Inc.). Expression of mRNA for interferon-gamma (*Ifng*), interleukin-4 (*Il4*), interleukin-22 (*Il22*)*,* the neutrophil attractant cytokine, keratinocyte-derived cytokine (*Kc*), transforming growth factor-beta (*Tgfb*), Toll-like receptor-4 (*Tlr4*), and tumor necrosis factor-alpha (*Tnfa*) were standardized against hypoxanthine-guanine phosphoribosyltransferase (*Hprt*), beta-glucuronidase (*Gusb*) and glyceraldehyde 3-phosphate dehydrogenase (*Gapdh*), as previously described [45]; these reference genes were selected due to the low variation among samples. A Quantstudio Real Time PCR instrument (Thermo Fisher Scientific) was used.

### 2.20. Quantification of Corticosterone

Serum and fecal corticosterone extractions were carried out according to the manufacturer’s instructions (Corticosterone Elisa Kit, Cayman Chemical, Ann Arbor, MI, USA). The optical densities of all corticosterone ELISAs (wavelength of 412 nm) were measured using a Synergy HT multi-detection microplate reader (BioTek Instruments Inc., Winooski, VT, USA) with Gen5 analysis software version 3.11 (BioTek Instruments Inc.).

### 2.21. Metabolomics

Kidney and liver tissues were homogenized in 4 mL/g methanol and 1.6 mL/g deionized water. Tissues were homogenized with a 6-mm-diameter steel bead for 5 min using a Qiagen TissueLyser LT at 50 Hz followed by 1 min of vortexing. This step was repeated two additional times to ensure complete tissue homogenization. To each sample, 2 mL/g chloroform was added and vortexed. Chloroform (2 mL/g) and deionized water (4 mL/g) were added to each sample and vortexed. Samples were then incubated at 4 °C for 15 min followed by centrifugation at 1000× *g* for 15 min at 4 °C. Next, 600 μL of the supernatant was removed and left until evaporated. Samples were rehydrated in 480 μL of metabolomics buffer (0.125 M KH_2_PO_4_, 0.5 M K_2_HPO_4_, 0.00375 M NaN_3_, and 0.375 M KF; pH 7.4). A 120 μL aliquot of deuterium oxide containing 0.05% *v*/*v* trimethylsilylpropanoic acid (TMSP) was added to each sample (total volume of 600 μL); TMSP was used as a zero-chemical-shift reference for ^1^H-NMR spectroscopy. A 550 μL aliquot was loaded into a 5-mm NMR tube and run on a 700 MHz Bruker Avance III HD spectrometer (Bruker Ltd., Milton, ON, Canada) for spectral collection. Data acquisition and processing were conducted as previously described [46].

### 2.22. Statistical Analyses

The majority of the statistical analyses were performed using Statistical Analysis Software version 9.4 (SAS Institute Inc., Cary, NC, USA). In conjunction with a significant main effect, the least-squares mean test was used to compare treatments within factors for bacterial densities, gene expression, and cytokine concentrations. The open field behavioral test results were analyzed using Student’s *t*-test. Fisher’s exact test was utilized to analyze categorical data (histopathological scoring and health assessments). For metabolomics data, NMR spectra were exported to MATLAB (Math Works) where they underwent spectral peak alignment and binning using Recursive Segment Wise Peak Alignment [47] and Dynamic Adaptive Binning [48], respectively. After these analyses, the dataset was normalized to the total metabolome excluding the region containing the water peak, which was log-transformed and Pareto scaled. MetaboanalystR was used to perform univariate and multivariate statistics including the calculation of fold changes of specific metabolites, heat-map creation, and hierarchical clustering analysis [49]. These tests were carried out using the bins identified as significant by univariate tests in order to observe group separation. Univariate measures include the *t*-test and the Mann–Whitney U test. Both tests determine if there is a significant difference between the means of the two groups; however, the *t*-test and the Mann–Whitney U test are applied in instances where the data are normally distributed (parametric) or not, respectively. The test for data normality was carried out using a decision-tree algorithm as described by Goodpaster et al. [50]. All *p*-values obtained from analysis were Bonferroni–Holm corrected for multiple comparisons. Metabolites were identified using Chenomx version 8.2 NMR Suite (Chenomx Inc., Edmonton, AB, Canada).

## 3. Results

### 3.1. Intestinal Colonization and Virulence Differed among EHEC O157:H7 Strains

At day 5 p.i., four of the EHEC O157:H7 strains evaluated (EDL933, TW14359, ECI-1375, and ECI-1911) negatively impacted (*p* < 0.001) the health of mice (Figure 1A). In contrast, the EHEC O157:H7 strain, FRIK-2001, did not affect the demeanor or appearance of mice. The impacts of EHEC O157:H7 strains EDL933, TW14359, ECI-1375 and ECI-1911 on mice necessitated that they were all humanely euthanized at the 5-day p.i. endpoint. Only EHEC O157:H7 negative mice (i.e., control treatment) and those inoculated with FRIK-2001 were allowed to continue to the target 10-day p.i. endpoint. However, the impacts of FRIK-2001 made it necessary to humanely euthanize mice at day 8 p.i.; at the 8-day p.i. endpoint, the FRIK-2001 strain resulted in an average health score of 2.0 ± 0.5, which was higher (*p* = 0.022) than control treatment mice not inoculated with EHEC O157:H7 (health score of 0.0) (Figure 1B). It is noteworthy that the maximum possible health score was eight, which included cumulative scores for general activity, hair-coat appearance and grooming behavior, and vocalization (Appendix A).

All five EHEC O157:H7 strains colonized mucosa in the ileum, cecum, proximal colon, and distal colon of mice at 5 days p.i. with no differences (*p* ≥ 0.200) in bacterial densities among the intestinal sites (Figure 2A–D). The degree of colonization was similar among the five EHEC O157:H7 strains at 5 days p.i., with the exception of FRIK-2001 and ECI-1375 in the ileum and distal colon where these strains were detected at a lower (*p* < 0.050) density than other strains. At all the intestinal locations examined, the mucosal colonization densities of FRIK-2001 between day 5 and day 8 p.i. remained the same (*p* ≥ 0.100). As the goal was to generate an enteric model of EHEC O157:H7 colonization that mimicked bovine, and strain FRIK-2001 was the only EHEC O157:H7 strain evaluated that did not incite acute disease within 5 days p.i. (presenting similarities to bovine intestinal colonization), this strain was selected for further experimentation (i.e., for the competitive colonization stress experiment).

### 3.2. All EHEC O157:H7 Strains Generated Histopathological Changes in the Intestine

Analysis of intestinal tissues revealed leukocyte infiltration, loss of goblet cells, crypt elongation and apoptosis in the intestines of all mice inoculated with EHEC O157:H7 strains. Specifically, the distal colon presented the highest degree of histopathologic changes relative to the control treatment, particularly for EHEC O157:H7 strains EDL933 and ECI-1911. Notably, stain FRIK-2001 presented lower histopathologic scores relative to EHEC O157:H7 strains EDL933 and ECI-1911 (*p* = 0.001 and *p* = 0.010, respectively) at 5 days p.i. (Figure 3). Relative to control treatment mice, modest histopathologic changes (*p* = 0.070) in the distal colon were observed in mice inoculated with FRIK-2001 at 8 days p.i.

### 3.3. Metabolite Profiles Were Altered in Kidneys of Mice Inoculated with EHEC O157:H7 Strains

The renal metabolome of mice inoculated with EHEC O157:H7 strains EDL933, TW14359, ECI-1375, and ECI-1911 was conspicuously changed (*p* < 0.001) relative to control treatment mice at 5 days p.i. (data not presented). In contrast, less conspicuous changes in the renal metabolome of mice inoculated with FRIK-2001 were observed at day 5 p.i., which did not differ from the renal metabolome of control treatment mice (Figure 4A,B). However, by 8 days p.i., differences were observed in the renal metabolome of mice inoculated with FRIK-2001 (193 altered bins) as compared to control treatment mice (Figure 4C,D). An examination of specific metabolites at 8 days p.i. revealed that concentrations of carnitine (*p* = 0.050) and kynurenine (*p* = 0.010) were decreased in mice inoculated with FRIK-2001 relative to control treatment mice.

### 3.4. Physiological Stress Affected the Behavior of Gnotobiotic Mice

Mice that were not administered corticosterone travelled at a faster rate during the exploration of the open field arena (*p* = 0.012) than mice administered corticosterone (i.e., stressed mice) (Figure 5A). In addition, the total distance travelled by mice not administered corticosterone was greater (*p* = 0.022) than by stressed mice (Figure 5B). No differences (*p* ≥ 0.680) in nest-building behavior by mice were observed among any of the treatments.

### 3.5. Corticosterone Concentrations Were Elevated in Mice Administered the Glucocorticoid

Regardless of the bacterial treatment, concentrations of corticosterone were higher in the serum (*p* = 0.006) and feces (*p* < 0.001) of stressed relative to non-stressed mice (Figure 6A,B). In addition, the hepatic metabolome in mice administered corticosterone was altered relative to mice that were not administered the stress hormone (Appendix A).

### 3.6. The Eighteen Commensal Escherichia coli Isolates from Cattle Were Genetically Distinct

The 18 commensal *E. coli* isolates evaluated were obtained from different animals, and SNP analysis showed that all isolates were phylogenetically distinct (Appendix A). Moreover, all commensal *E. coli* strains produced distinct PFGE profiles (data not presented), and analysis of WGS data showed that the strains belonged to distinct serotypes, phylotypes (B1 was the main phylogroup containing 11 out of the 18 commensal *E. coli* strains examined), and/or MLST subtypes (Appendix A). Analysis of WGS data identified carriage of antimicrobial resistance determinants in two (i.e., strain 1309 and strain 1310) of the eighteen commensal *E. coli* strains, and eleven of the strains carried at least one plasmid (Appendix A). All of the commensal *E. coli* strains carried an array of putative virulence factors, but none of the strains carried pathogenicity/virulence genes linked to hemorrhagic colitis and HUS (e.g., *stx*) (Appendix A), which is consistent with our observation of no measureable Stx production by any of the 18 commensal *E. coli* strains. Interestingly, only one of the commensal *E. coli* strains possessed genes encoding bacteriocins (i.e., strain 1315); this strain carried genes predicted to encode colicins/microcins, including *cba* (colicin B), *cib* (colicin Ib), *cma* (colicin M), *mchB* (microcin H47), and *mcmA* (microcin M). Furthermore, *E. coli* strain 1315 is the only commensal strain that belonged to phylogroup B2, and possessed genes encoding the iron-acquisition systems, yersiniabactin (*fyuA*) and salmochelin (*iroN*), as well as the colibactin genotoxin (*clb*/*pks* island).

### 3.7. DNA of Commensal Escherichia coli Strains Propagated Separately or Communally Were Detected in the Intestines of Mice

Commensal *E. coli* isolates were assigned the identifiers 1303 through 1322 (excluding 1311 and 1320), and primers were developed and used to detect and quantify individual strains in the distal colon of mice. In addition to targeting individual strains, total quantities of *E. coli* were quantified. There was no difference (*p* ≥ 0.430) in total densities of *E. coli* in the distal colon of mice that were administered commensal strains that were grown separately or communally, nor between stressed and unstressed mice (Appendix A). Unexpectedly, densities of total *E. coli* were lower (*p* = 0.036), albeit marginally, in mice that were administered EHEC O157:H7 (i.e., FRIK-2001). Density measurements of individual *E. coli* strains showed that all 18 strains were present in the distal colon (Appendix A). Densities differed (*p* ≤ 0.054) among strains, and strains 1308, 1315, and 1318 were consistently detected at lower (*p* ≤ 0.050) densities. For the majority of the strains, there were no conspicuous differences (*p* ≥ 0.058) in the densities of individual commensal *E. coli* strains between isolates grown separately or communally, between stress or unstressed mice, nor between mice with or without EHEC O157:H7. However, densities in the colon differed (*p* ≤ 0.051) for *E. coli* strains 1312, 1313, 1315 that were grown communally vs. separately, and densities also differed (*p* ≤ 0.036) for *E. coli* strains 1318 and 1319 in animals that were inoculated with EHEC O157:H7. Consistent with the detection of DNA of all 18 commensal *E. coli* strains in the distal colon of mice administered the bacteria, all of the strains were also detected in the culture medium (i.e., for isolates grown communally). In this regard, 300 arbitrarily selected isolates were characterized by PFGE, and frequencies of recovered strains ranged from 1.1 to 13.2% (Appendix A).

### 3.8. Density of EHEC O157:H7 Was Reduced in Mice Administered Commensal Escherichia coli

Cell densities of FRIK-2001 associated with cecal mucosa were lower in mice inoculated with commensal strains produced communally (CC, *p* = 0.017) and separately (CS, *p* < 0.001) (Figure 7A). The same response was observed in the proximal colon (EHEC + CC, *p* = 0.015; EHEC + CS, *p* = 0.009) and distal colon (EHEC + CC, *p* = 0.057; EHEC + CS; *p* = 0.009) (Figure 7B,C). Furthermore, the administration of commensal *E. coli* strains reduced densities of FRIK-2001 in digesta within the ileum (EHEC + CC, *p* < 0.010; EHEC + CS, *p* < 0.001), cecum (EHEC + CC, *p* = 0.001; EHEC + CS, *p* < 0.001), proximal colon (EHEC + CC, *p* = 0.002; EHEC + CS, *p* < 0.001), and distal colon (EHEC + CC, *p* = 0.004; EHEC + CS, *p* = 0.003) (Appendix A). In no instance was FRIK-2001 detected in mice not inoculated with the pathogen. Stress did not alter (*p* ≥ 0.118) the densities of EHEC O157:H7. There was no difference (*p* ≥ 0.196) in EHEC O157:H7 densities in digesta or associated with mucosa in mice administered *E. coli* strains grown communally or separately.

### 3.9. Histopathologic Changes Incited by EHEC O157:H7 Were Most Severe in the Distal Colon

Consistent with experiment one, mice inoculated with FRIK-2001 exhibited no evidence of clinical illness at 3 days p.i. However, modest histopathologic changes in mice inoculated with the pathogen were observed throughout the intestinal tract, and more extensive histopathologic changes were observed in the distal colon relative to the cecum (*p* < 0.001) and proximal colon (*p* < 0.001) (Figure 8A). Stress, alone or interactively with EHEC O157:H7, did not affect histopathologic scores.

### 3.10. Histopathologic Changes Incited by EHEC O157:H7 in the Distal Colon Were Reduced in Mice Administered Commensal Escherichia coli

No deleterious impacts of the commensal *E. coli* stains alone were observed (i.e., CC and CS vs. control treatment) (Figure 8B). Both of the commensal *E. coli* treatments (i.e., EHEC + CC and EHEC + CS) reduced (*p* ≤ 0.003) histopathologic changes in the distal colon of mice infected with FRIK-2001 (i.e., EHEC treatment), equal to mice not inoculated with the pathogen. There was no effect (*p* = 0.100) of stress induction on histopathologic scores.

### 3.11. Expression of Inflammatory Markers in the Distal Colon Was Reduced in Mice Administered Commensal Escherichia coli

Expression of *Tnfa* and *Kc* mRNA was reduced (*p* < 0.001) in mice administered commensal *E. coli* as compared to mice administered EHEC O157:H7 alone (Figure 9A,B). Commensal *E. coli* did not affect (*p* ≥ 0.124) quantities of *Il4*, *Il22*, *Tlr4*, or *Ifng* mRNA. *Tnfa* expression was higher (*p* = 0.018) in mice administered corticosterone and FRIK-2001 alone (Figure 9A). Moreover, corticosterone administration reduced (*p* < 0.001) the mRNA concentration *Tgfb* independent of EHEC O157:H7 and commensal *E. coli* administration (Figure 9C).

## 4. Discussion

### 4.1. Development of a Bovine EHEC O157:H7 Intestinal Colonization Model

Due to the complex nature of the microbiota present in the intestinal tract of individual cattle, it is challenging to elucidate interactions among bacteria, and their impacts on the host. Therefore, a simpler enteric model devoid of a microbiota would facilitate the assessment of bacterial competition and bacteria–host interaction toward the understanding of mechanisms and the formulation of hypotheses for subsequent validation in cattle. Colonization mechanisms in the intestine, shedding patterns, and lack of EHEC O157:H7 disease in cattle are factors that are still not fully understood, and further information is needed to enable the development of on-farm mitigation strategies. The development and use of a GB murine model of enteric bovine EHEC O157:H7 colonization could provide key information on essential aspects of the pathogen–host–microbiota interaction. Therefore, an objective of the current study was to develop a murine GB EHEC O157:H7 colonization model that possesses similarities with cattle colonized by the pathogen. In particular, the characteristics of such a model could potentially resemble the undeveloped intestinal tract of the young calf, which at a young age has a digestive system that functions similarly to a monogastric animal [51]. EHEC O157:H7 mouse models have mainly been used as a human model to study HUS [23,24,52]. In this regard, both GB and streptomycin-treated murine models have been utilized due to their susceptibility to Stx with a resultant development of renal damage and death [23,24,42]. Initially, we examined the colonization and health status of mice inoculated with five different EHEC O157:H7 strains, representing different phylogenic lineages [53,54,55,56]. All five strains successfully colonized mucosa in the large intestine (cecum, and proximal and distal colon) at 5 days p.i. with no differences in bacterial densities among the intestinal sites. However, FRIK-2001 was the only strain that did not present elevated health scores before day 5 p.i. In contrast, mice inoculated with either EDL933, TW14359, ECI-1375, or ECI-1911 presented sufficiently high scores that they had to be humanely euthanized on day 5 p.i. (i.e., observations beyond 5 days p.i. were not possible). FRIK-2001 is a lineage II EHEC O157:H7 from bovids [54], and mice inoculated with this strain displayed a reduction in general activity and in hair-coat appearance only at 8 days p.i. Therefore, FRIK-2001 was capable of colonizing the intestinal tract of GB mice with delayed symptomatology. In susceptible mice infected with EHEC O157:H7, there is typically a rapid progression from colitis to renal failure where Stx produced by EHEC O157:H7 causes acute tubular necrosis that can lead to death [23,42]. Therefore, the rapid and severe symptomatology observed in mice inoculated with either EDL933, TW14359, ECI-1375 or ECI-1911 could be intimately related to kidney failure. It is noteworthy that in contrast to FRIK-2001, EHEC O157:H7 strains EDL933, TW14359, ECI-1375, and ECI-1911 belong to lineage I or I/II. Moreover, lineage II EHEC O157:H7 strains produced less Stx1 and Stx2 than lineage I or I/II strains [56], which may explain the reduced impacts of FRIK-2001 on mice that were observed in the current study.

Mice infected with FRIK-2001 did not develop diarrhea; however, the bacterium triggered moderate leukocyte infiltration, goblet-cell loss, crypt hyperplasia, and apoptosis of epithelial cells by 8 days p.i. The largest histopathologic changes were observed in the distal colon of mice, much like in cattle where the distal colon is the location where attaching effacing lesions occur [57]. Other EHEC O157:H7 strains examined such as EDL933 and ECI-1911 incited significantly higher histopathologic changes at day 5 p.i. than FRIK-2001. Notably, inflammation in mice infected with FRIK-2001 at day 5 p.i. was categorized as mild, and was reduced in the distal colon by day 8 p.i. Previous studies utilizing strain EDL933 in GB mice have categorized the colonic alterations as a necrotizing colitis accompanied with a few attaching effacing lesions [42]. Similarly to our findings, Eaton et al. [42] found that necrotizing colitis peaked at 1–4 days p.i. and was gradually reduced thereafter. We observed substantive metabolomic changes in the kidneys of mice infected with the EHEC O157:H7 strains evaluated. Significantly, renal metabolomic profiles of mice infected with EHEC O157:H7 strains corresponded to health-assessment scores. In this regard, mice infected with FRIK-2001 exhibited significantly altered renal metabolite bins at day 8 p.i. but not at day 5 p.i. The lack of metabolomic differences between FRIK-2001-inoculated mice and control treatment mice at 5 days p.i. supports our conclusion that early intestinal colonization by this EHEC O157:H7 strain occurs in the absence of clinical kidney pathology. Carnitine, a biologically active metabolite, was found to have a 25%-fold decrease in mice infected with FRIK-2001 at 8 days p.i. as compared to control treatment mice. Carnitine is involved in the transportation of free fatty acids into the mitochondria for β-oxidation [58]. Notably, the presence of Stx2a plus tumor necrosis factor-alpha (TNFα) reduced carnitine in human renal glomerular endothelial cells (HRGEC) [59]. Low levels of carnitine show a perturbation in the normal use of free fatty acids [59], which suggests that an alteration in normal metabolism occurs due to kidney disruption by 8 days p.i. The metabolite kynurenine was also significantly reduced in mice infected with FRIK-2001 at 8 days p.i. Elevated levels of kynurenine occur in HRGEC following the combined administration of Stx2a and TNFα [59]. However, the role of kynurenine is complex since elevated levels of kynurenic acid (i.e., a metabolite of kynurenine) can act as an early mediator of leukocyte recruitment [60]. Nonetheless, kynurenic acid is also capable of reducing lipopolysaccharide (LPS)-stimulated secretion of interferon-gamma (IFNγ) and TNFα [61]. This indicates a downregulating role of some factors of the immune response. Given the lower amounts of kynurenine that we observed in kidneys, it is possible that an alteration in the regulation of certain aspects of the immune response was taking place at 8 days p.i. Intestinal colonization, histopathologic changes, metabolomic profiles, and health-assessment scores at day 5 and 8 p.i. indicated that FRIK-2001 more closely mimics colonization in bovine relative to other EHEC O157:H7 strains. Notably, lineage II strains exhibit a unique host ecology relative to lineage I and I/II EHEC O157:H7 strains [56,62], and they are more commonly isolated from bovine hosts, whereas lineage I strains are more frequently isolated from human beings [62]. It is noteworthy that the aim of the current research was to develop an intestinal model that allows researchers to mechanistically examine the enteric competition and behavior of EHEC O157:H7 strains of bovine origin and develop potential mitigations. It is not meant to replace cattle, but rather to facilitate validation/evaluations in cattle.

### 4.2. Induction of Physiological Stress in Mice Had a Minimal Impact on EHEC O157:H7 Intestinal Colonization and Pathology

The GB murine model allowed us to examine the impact of physiological stress on host–bacterial interactions. To mimic the physiological consequences of stress, we administered the glucocorticoid stress hormone, corticosterone, in drinking water. The hepatic metabolome of mice administered corticosterone was altered, as has been reported previously [63,64]. To further evaluate the impact of stress, the behavior of mice was assessed with an open field test and nest-building ability. The open field test enables a reliable assessment of mouse exploratory drive (curiosity) as well as anxiety (fear), which can be directly affected under stressful scenarios [65]. Similarly to others using normal flora mice models, we observed that GF and GB mice administered corticosterone presented a reduction in total locomotion as well as a slower travelling rate than non-stressed mice. This indicates that stressed GF and GB mice exhibit reduced exploratory behavior when encountering a new open and unprotected environment [35,37]. Although others have reported that corticosterone-treated mice exhibit a significant reduction in the time spent in the center of an open field [66], we did not observe this behavior. It has been suggested that GF mice show reduced anxiety-like behavior (anxiolytic) as compared to normal flora mice when placed in the elevated plus maze, indicating that the microbiota plays a role in the development of anxiety in a mouse [67]. Our mice were GF or GB, and thus the differences we observed in anxiety-like behavior could be attributed directly to physiological stress induced by corticosterone. This was supported by the higher levels of corticosterone in the serum of stress-treatment mice. In contrast, we did not observe that corticosterone administration influenced the nest-building ability of the mice. This could be due to mice only being stressed for 9 days before the test was implemented. It is noteworthy that maintaining the GF and GB status of mice represents challenges in conducting behavioral analyses, which necessitated that we could only conduct behavioral analysis using in-cage assessments (e.g., nest building), and other classical behavioral assessments, at the end of the experimental period (i.e., immediately before euthanization).

Previous studies have suggested that stress may be a predisposing factor benefiting colonization of the intestine of cattle by EHEC O157:H7 [68]. In our model, we did not find that a general state of stress benefitted intestinal colonization by FRIK-2001. Previous observations have shown that stress hormones, such as catecholamines (epinephrine and norepinephrine) affect the expression of virulence factors by EHEC O157:H7 [68,69,70]. These molecules were found to enhance the expression of the type three secretion system, a complex specifically targeted towards binding with epithelial cells and forming attaching effacing lesions [68,69,70]. Additionally, in vitro studies demonstrated that a direct interaction with catecholamines stimulated the growth of *E. coli* strains [71]. In the current study, stress did not enhance the densities of FRIK-2001 or intestinal damage caused by the bacterium. It is noteworthy that mice were physiologically stressed with corticosterone, a known systemic glucocorticoid hormone released in mice under stressful situations [72]. Glucocorticoids have been shown to stimulate the production of catecholamines [73]. We do not know if corticosterone stimulated the secretion of catecholamines in the intestine of the mice; the influence of catecholamine on EHEC O157:H7 is difficult to replicate in vivo, and has only previously been observed in vitro or in ligated intestinal loops [68,69].

We observed an impact of stress in the expression of *Tnfa* and *Tgfb* in the distal colon of GB mice. Regulatory molecules such as TGFβ have the important function of controlling the inflammatory response, and avoiding collateral damage to host tissue [74]. The reduction in expression of this cytokine in the distal colon could lead to dysregulation of the inflammatory response and thereby benefit EHEC O157:H7 colonization. Moreover, EHEC O157:H7 infection has been shown to cause reduced protein expression in tight junctions and barrier dysfunction [75]. TGFβ is capable of preventing epithelial-barrier dysfunction generated by EHEC O157:H7, and potentially reducing the penetration of Stx past the epithelium [75]. The reduction of *Tgfb* expression in the distal colon that we observed in stressed animals could possibly play a pivotal role in the colonization of FRIK-2001. Stressed mice mono-colonized with FRIK-2001 presented significantly elevated levels of *Tnfa* in the distal colon as compared to non-stressed mice infected with FRIK-2001. TNFα stimulates a downstream cascade with the consequent arrival of neutrophils to the lamina propria of the intestine [76,77]. Neutrophils represent a first line of defense in response to EHEC O157:H7, and therefore are of importance in the ability of the host to eliminate bacterial pathogens [78]. However, TNFα can possibly benefit EHEC O157:H7, as the administration of a TNFα inhibitor reduced pathologic effects in mice infected with EHEC O157:H7 [79].

### 4.3. Commensal Escherichia coli Strains of Bovine Origin Administered to Mice Reduced EHEC O157:H7 Colonization and Pathology

Effective mitigation of EHEC O157:H7 in cattle is challenging, and the development of successful strategies remains elusive. Multiple methods have been employed in an attempt to eliminate EHEC O157:H7 in cattle, its primary reservoir, many of which include pre-harvest strategies that attempt to reduce exposure of cattle to the pathogen or exclude it from the gastrointestinal tract and hides [80]. In the current study, a combination of 18 *E. coli* commensal strains of bovine origin significantly reduced densities of FRIK-2001 in the intestinal tract of mice, particularly in the distal colon. Previous studies have utilized murine models to ascertain the impact of human commensal *E. coli* strains on EHEC O157:H7 with the goal of eliminating EHEC O157:H7 from the intestine of afflicted people [17,21,81]. In these studies, the commensal *E. coli* strains were effective at reducing EHEC O157:H7 colonization [17]; however, a dysbiosis was first achieved with streptomycin, and mice were pre-colonized with the commensal strains 10 days prior to inoculation with EHEC O157:H7. It is noteworthy that the adhesion of *E. coli* strains to the epithelium can provide an advantage for strains pre-colonizing the intestine [82]. In contrast to human-focused studies, we inoculated GF mice with FRIK-2001 and commensal *E. coli* strains at the same time with the idea of precluding a pre-colonization advantage to the commensal competitors. Freter et al. [82] proposed that the competition for colonization niches could be, in part, based on competition for nutrients. Different strains of *E. coli* have different nutrient requirements [81], and the inability to utilize a nutrient at a higher rate than other bacteria can be a limiting survival factor in the intestinal environment. In a previous study, a single commensal *E. coli* strain did not outcompete EHEC O157:H7 [17]. This suggests that a variety of commensal *E. coli* strains are required; however, we did not ascertain to what degree individual or combinations of individual strains were effective, which was beyond the scope of the current study. Regardless, one or all of the 18 commensal *E. coli* strains administered were able to inhibit FRIK-2001, likely by affecting its ability to access key nutrients, thus reducing its survival. Given that all of the administered commensal strains of *E. coli* colonized the intestine of GB mice suggests that they occupy different niches. It is plausible that the presence of multiple competitive commensal *E. coli* strains in different niches collectively reduced the survival and proliferation rate of EHEC O157:H7, and the mechanisms involved warrant additional investigation. Alternatively, certain *E. coli* can produce bacteriocins (e.g., colicins/microcins) that can have a direct effect on EHEC O157:H7 [83]. Analysis of WGS data of the commensal *E. coli* strains utilized in the current study revealed that only a single strain (i.e., strain 1315) carried genes encoding colicin/microcin bacteriocins. Furthermore, *E. coli* strain 1315 possessed similarities to the commonly used probiotic *E. coli* strain Nissle 1917, such as being a B2 phylotype, and carriage of the iron-acquisition systems, yersiniabactin- (*fyuA*) and salmochelin- (*iroN*), and colibactin genotoxin- (*clb*/*pks* island) and microcin M/H47-encoding genes. Importantly, *E. coli* strain Nissle 1917 has previously been shown to reduce EHEC O157:H7 colonization in mice [81]. It is plausible that the secretion of colicins and/or microcins played a role in the observed exclusion of EHEC O157:H7. These bacteriocins can produce a pore in bacterial cell membranes, can degrade peptidoglycan precursors, have phosphatase activity, and target 16S RNA and DNAse activity [84]. Evidence of bacteriocin antimicrobial activity against different pathogenic *E. coli*, including EHEC O157:H7, has previously been demonstrated both in vitro and in vivo [83,85,86,87].

We have previously observed that some commensal *E. coli* isolates increased the amount of toxin generated by a high-toxoid-producing EHEC O157:H7 strain in vitro (Gannon unpublished data). Furthermore, commensal *E. coli* strains can lysogenize with a Stx2-converting phage and produce Stx2 in vitro [83]. Thus, the evaluation of commensal *E. coli* strains in vivo using an appropriate model, such as the GB mouse model described herein, will inform the selection of appropriate strains toward competitive exclusion evaluations in cattle.

Histopathological changes such as inflammatory infiltrates in the mucosa, apoptosis of enteric cells, epithelial hyperplasia, cryptitis, and goblet-cell loss were observed to be the highest in the distal colon of GB mice in the current study. We were particularly interested in examining histopathologic changes in the distal colon as this area of the intestine, and specifically the recto–anal junction (RAJ), is considered to be the main site of EHEC O157:H7 colonization in cattle [57,84]. We observed that the presence of commensal *E. coli* significantly reduced histopathologic scores in the distal colon of mice, suggesting that one or more of the 18 commensal *E. coli* strains administered interfered with colonization and subsequent epithelial damage incited by FRIK-2001. Importantly, the histopathologic changes that we observed in mice mono-colonized with FRIK-2001 correspond with histopathologic changes previously described in cattle [85,86,87]. In this regard, calves colonized with EHEC O157:H7 develop a mild neutrophilic inflammation in the mucosa of the large intestine [87], an alteration that we observed in the mucosa layer of the distal colon of GB mice.

Relative expression of *Tnfa* and *Kc* in the distal colon was significantly reduced in GB mice colonized by commensal *E. coli* strains, indicating that they ameliorated the inflammatory impact of EHEC O157:H7. KC is a chemokine that shares functional properties with human interleukin-8 (IL8), causing a strong neutrophil attraction to the site of inflammation [88]. KC has been linked with the functional role of inducing neutrophil accumulation in the glomeruli of the kidney of mice when stimulated with *E. coli* LPS or Stx2 [88]. The expression of this chemotactic chemokine is of particular importance as accumulation of neutrophils are observed in the colonic mucosa of calves infected by EHEC O157:H7 [85]. This suggests that neutrophils are important in the clearance of EHEC O157:H7 from the intestinal tract of cattle. The increase in *Kc* expression observed in the distal colon is consistent with the neutrophilic infiltration that we observed in GB mice infected with EHEC O157:H7. However, other studies reported a reduction in the transcription of pro-inflammatory immune factors in the RAJ of super-shedding cattle, implicating the decrease in transcription of inflammatory cytokines as a possible factor in EHEC O157:H7 colonizing the RAJ [89]. We observed that *Tnfa* was also reduced in GB mice colonized by commensal *E. coli* strains. TNFα is a pro-inflammatory cytokine known to trigger the activation of several elements of the immune response including the transcription factor NF-κβ with the consequent activation of inflammatory signalling pathways and pro-inflammatory cytokines such as IL8 [77,90]. Inhibition of TNFα in mice infected with EHEC O157:H7 was associated with reduced pathology and animal lethality [79]. Furthermore, the infection of mice with EHEC O157:H7 induces the secretion of NF-κβ and consequently IL8 [91]. It is noteworthy that infection by EHEC O157:H7 Stx-positive strains induces greater NF-κβ expression than infection by EHEC O157:H7 Stx-negative strains [91]. Although EHEC O157:H7 infection elevates the secretion of TNFα, infection by EHEC O157:H7 can also reduce the activation of NF-κβ; however, the mechanism is not fully understood [92]. Competition with commensal *E. coli* may have disrupted the ability of FRIK-2001 to interact with intestinal mucosa, thereby reducing host recognition and subsequently the expression of inflammatory markers. Notably, the reduced expression of *Tnfa* in mice inoculated with commensal *E. coli* that we observed corresponded with reduced intestinal damage. In contrast to mice, however, adolescent cattle inoculated with EHEC O157:H7 do not exhibit an elevated expression of *Tnfa* [93]. Overall, our findings indicated that competition from commensal *E. coli* strains reduced FRIK-2001 densities, lessened histopathologic lesions, and decreased pro-inflammatory markers. It is plausible that the commensal *E. coli* affected the ability of FRIK-2001 to access nutrients, thereby reducing densities of the pathogen. Furthermore, interfering with direct access to the epithelium would be expected to reduce recognition of the pathogen by the host, thereby lowering activation of pro-inflammatory markers and ensuing damage to the host’s epithelium. As one of the commensal *E. coli* strains that we evaluated carried genes that encode colicins/microcins, it is possible that the production of bacteriocins may have also contributed to the mitigation of EHEC strain, FRIK-2001.

### 4.4. Growing Commensal Escherichia coli Strains Communally Did Not Increase Efficacy

There was no observable difference on the efficacy of EHEC O157:H7 exclusion between commensal *E. coli* strains grown separately or communally. Previous studies reported success in reducing infection of chickens by *S. enterica* serovar Typhimurium via incubating mixtures of cultures of commensal bacteria before competition with the pathogen [63,64]. The reason for the enhanced efficacy of cultures grown communally is currently unknown. A possible explanation is that competition amongst commensal bacteria enhances glycocalyx formation, providing a survival advantage later in the intestine [64]. Another possibility is that communal growth triggers quorum sensing (QS) among bacteria, where signals can be used to synchronize behaviors of the population and prompt bacteria to act as multicellular organisms [65]. Some bacteria have the ability to quorum quench, where they inactivate signals from other bacteria that interfere with communal communication [65]. At high bacterial numbers, enough molecules are produced for communal detection; however, QS has not been studied under this specific scenario. It is noteworthy that communal incubations of bacteria for use against *S. enterica* serovar Typhimurium were conducted for 7, 24, and 48 h, and the latter two times were found to be the most effective [63,66]. It is possible that prolonged incubation periods (i.e., beyond log phase) may enhance the competitive efficacy of commensal *E. coli* strains grown communally.

## 5. Conclusions

We developed a GB mouse model to study the host-EHEC O157:H7 interaction in a manner that mimics the interaction in cattle. The advantage of this model is the absence of an enteric microbiota that aids in the elucidation of mechanisms (i.e., in the absence of the confounding effects of the enteric microbiota). Contrary to our hypothesis, the administration of exogenous corticosterone as a model of general physiological stress did not facilitate colonization of the intestinal tract by EHEC O157:H7. However, the administration of 18 commensal *E. coli* strains at the same time as EHEC O157:H7 effectively reduced densities of the pathogen and pathologic impacts on the host. Importantly, the murine model developed in the current study can be used to elucidate mechanisms of pathogenesis and colonization resistance toward the development of effective on-farm mitigations for EHEC O157:H7 in cattle.

## Figures and Tables

**Figure 1 animals-13-02577-f001:**
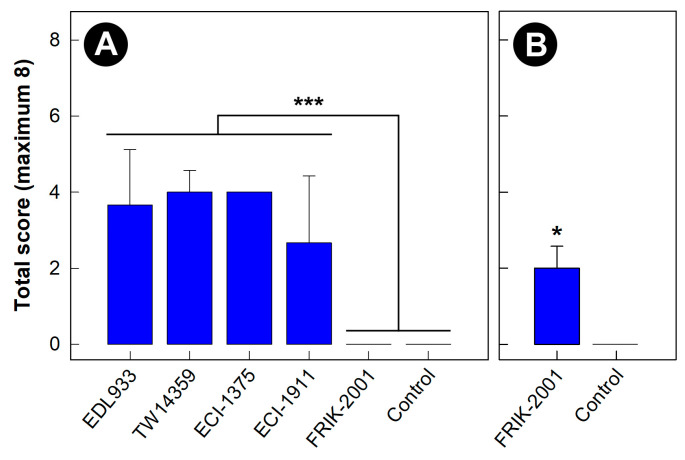
Health assessment scores of mice inoculated with one of five enterohemorrhagic *Escherichia coli* O157:H7 strains or phosphate buffer alone (Control). Scoring parameters: general activity (0–4), hair-coat appearance and grooming behavior (0–3), and vocalization (0–1). (**A**) At 5 days post-inoculation. (**B**) At 8 days post-inoculation. Mice were inoculated with one strain. Histogram bars indicated with an asterisk differ (* *p* = 0.022, *** *p* < 0.001, *n* = three replicate mice per treatment per time post-inoculation).

**Figure 2 animals-13-02577-f002:**
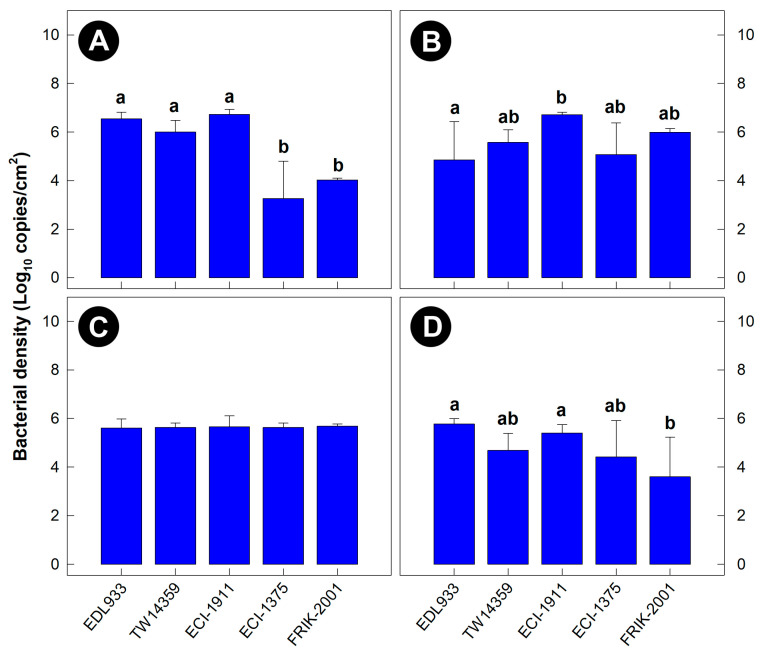
Densities of enterohemorrhagic *Escherichia coli* O157:H7 cells associated with intestinal mucosa at 5 days post-inoculation. (**A**) Ileum. (**B**) Cecum. (**C**) Proximal colon. (**D**) Distal colon. Histogram bars not indicated with the same letter differ (*p* ≤ 0.050, *n* = three replicate mice per treatment).

**Figure 3 animals-13-02577-f003:**
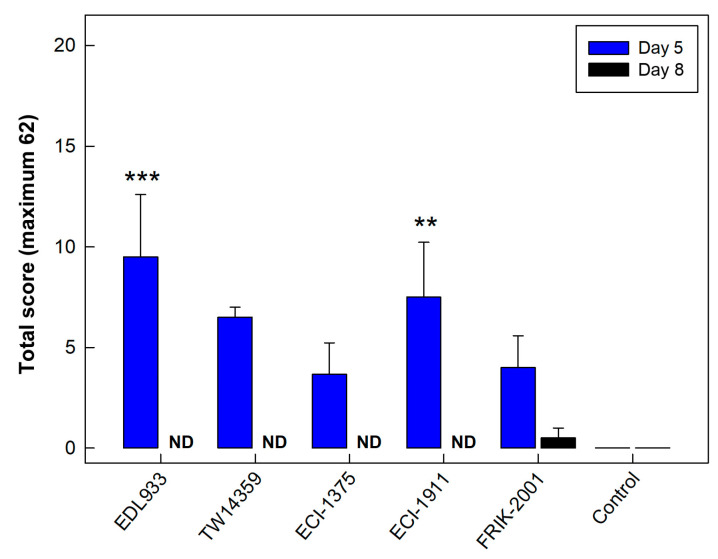
Total histopathological score in the distal colon of mice inoculated with one of five enterohemorrhagic *Escherichia coli* (EHEC) O157:H7 strains or phosphate buffer alone (Control). Mice were evaluated at 5 and 8 days post-inoculation. Mice inoculated with EHEC O157:H7 strains, EDL933 and ECI-1911 exhibited higher total score than the EHEC O157:H7 strain, FRIK-2001, where ** is *p* = 0.010 and *** is *p* = 0.001 (*n* = three replicate mice per treatment). Although not significantly different (*p* = 0.070), a trend for a lower histopathologic change score was observed in mice inoculated with FRIK-2001 at the day 8 endpoint relative to control treatment mice. Due to the detrimental impacts of EHEC O157:H7 strains, EDL933, TW14359, ECI-1375, and ECI-1911 on mice, it was not possible to conduct histopathologic examinations at the day 8 endpoint (ND, not determined).

**Figure 4 animals-13-02577-f004:**
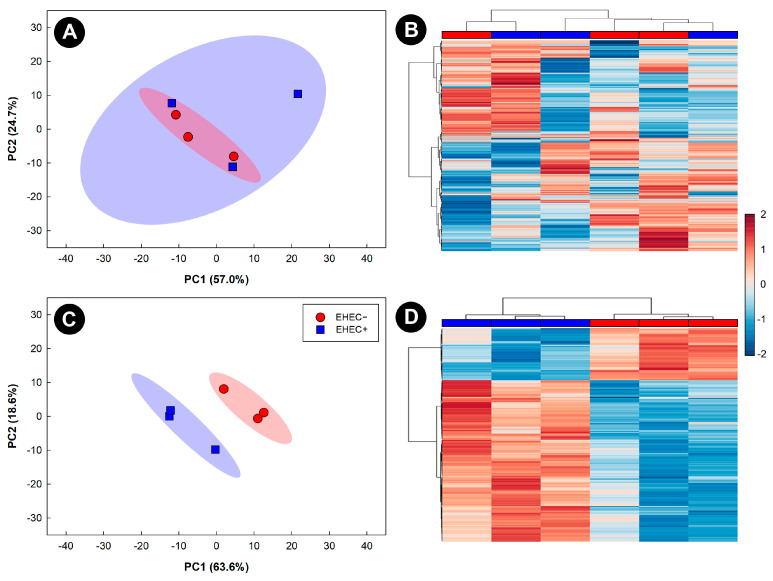
Principal component analysis (PCA) score plots and heat maps of metabolite bins in kidneys of mice inoculated with the enterohemorrhagic *Escherichia coli* O157:H7 (EHEC) strain, FRIK-2001 (EHEC+; blue markers and bars) or administered with phosphate-buffered saline alone (EHEC–; red markers and bars). (**A**) PCA score plot at 5 days post-inoculation (p.i.). (**B**) Heat map at 5 days p.i. (**C**) PCA score plot at 8 days p.i. (**D**). Heat map at 8 days p.i. For the PCA score plots, each marker represents one mouse, and the colored ellipses represent 95% confidence intervals. The x- and y-axes show principal components one and two, respectively, with the number in brackets indicating the percent variance of each component. Bins were determined to differ (*p* ≤ 0.050) according to a univariate Mann–Whitney U-test. For the heat maps, the degree to which metabolite bins were up- or down-regulated is denoted with red and blue coloration, respectively. The dendrogram associated with the heat map illustrates the results of the unsupervised hierarchical clustering analysis.

**Figure 5 animals-13-02577-f005:**
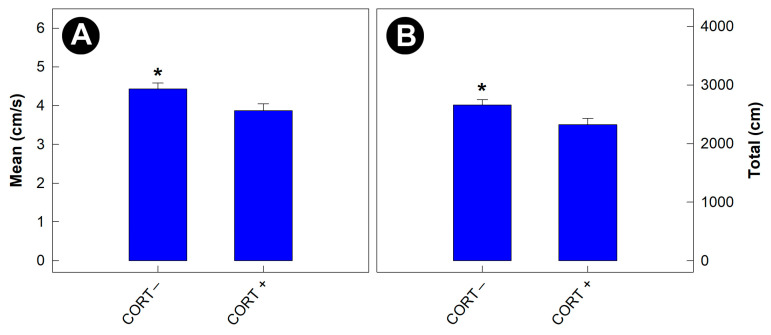
Results of a 10 min open field test of mice administered corticosterone in drinking water (CORT+) or drinking water without corticosterone (CORT–). Corticosterone was administered for 9 consecutive days. (**A**) Mean velocity during the entire 10 min open field test. The asterisk denotes that the mean velocity differed between the treatments (*p* = 0.012). (**B**) Total distance travelled in a 10 min open field test. The asterisk denotes that the mean velocity differed between the treatments (*p* = 0.022). As there was no impact of enterohemorrhagic *Escherichia coli* O157:H7 or commensal *E. coli* treatments, data were averaged over these factors.

**Figure 6 animals-13-02577-f006:**
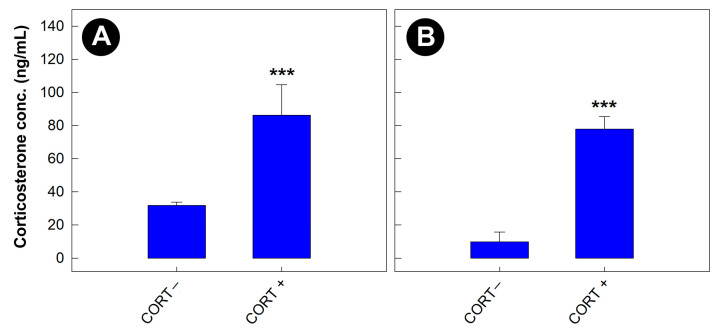
Corticosterone concentrations in mice administered corticosterone in drinking water (CORT+) or drinking water without corticosterone (CORT–). Corticosterone was administered for 9 consecutive days. (**A**) Serum. (**B**) Feces. Histogram bars indicated with an asterisk differ (*** *p* < 0.001). As there was no impact of enterohemorrhagic *Escherichia coli* O157:H7 or commensal *E. coli* treatments, data were averaged over these factors.

**Figure 7 animals-13-02577-f007:**
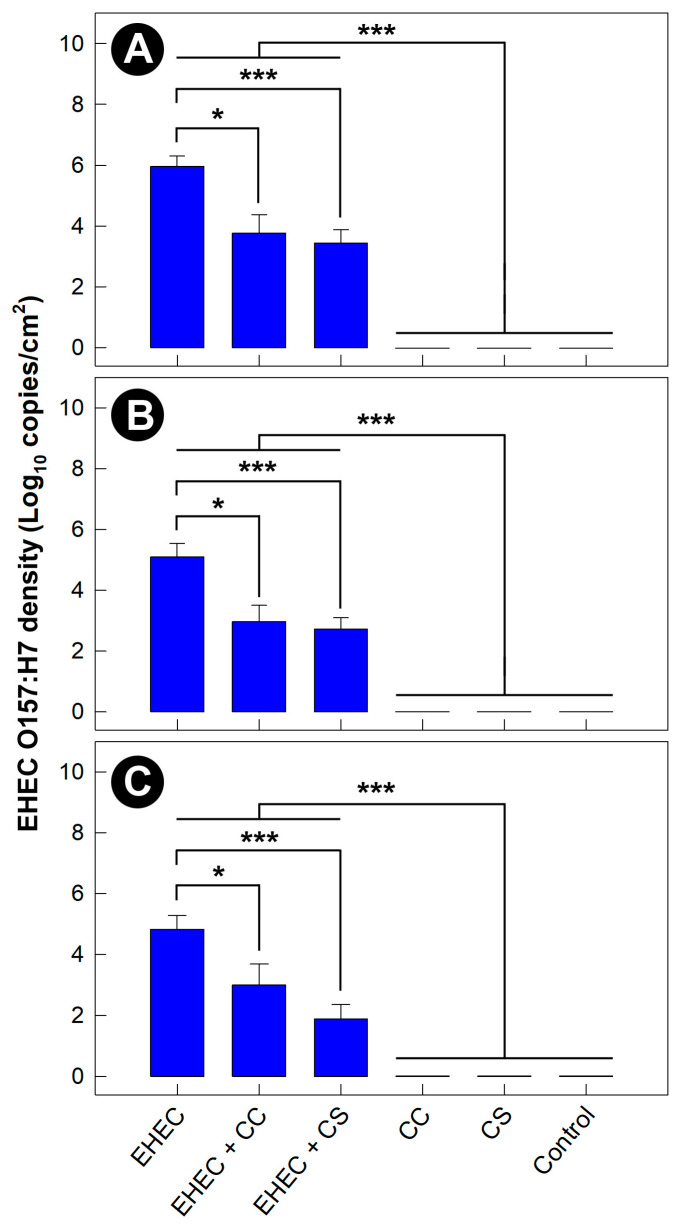
Densities of the enterohemorrhagic *Escherichia coli* O157:H7 (EHEC) strain FRIK-2001 associated with intestinal mucosa. (**A**) Cecum. (**B**) Proximal colon. (**C**) Distal colon. EHEC densities were determined using qPCR with primers that were specific for FRIK-2001. Histogram bars indicated with asterisks differ (* *p* < 0.050; *** *p* < 0.001). As there was no impact of corticosterone on densities of EHEC, data were averaged over the corticosterone treatment.

**Figure 8 animals-13-02577-f008:**
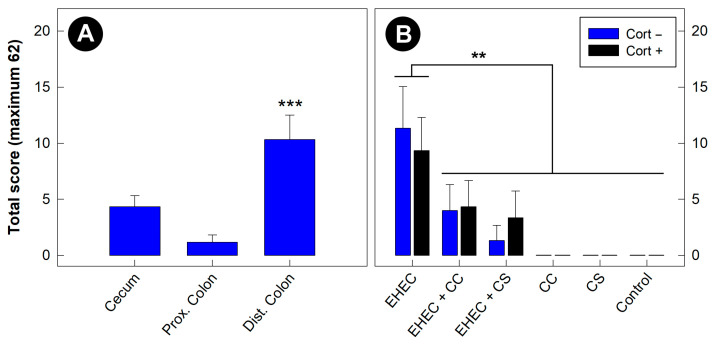
Total histopathologic change scores in mice administered enterohemorrhagic *Escherichia coli* O157:H7 (EHEC) strain FRIK-2001. (**A**) Histopathologic scores in the large intestine of mice administered EHEC alone. Scores in distal colon were higher (*p* < 0.001) relative to other locations. (**B**) Histopathologic scores among treatments in the distal colon. Scores for mice administered EHEC alone were higher (*p* ≤ 0.003) than for other treatments. As there was no impact of corticosterone on histopathologic change scores among treatments, data were averaged over the corticosterone treatment. Histogram bars indicated with asterisks differ (** *p* < 0.010; *** *p* < 0.001).

**Figure 9 animals-13-02577-f009:**
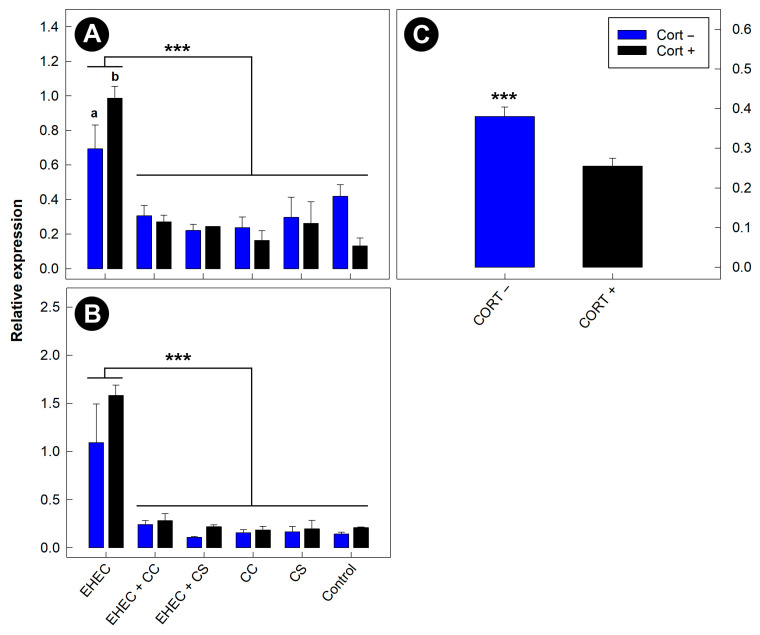
Relative expression of mRNA expression in the distal colon of mice. (**A**) Tumor necrosis factor-alpha (*Tnfa*). For mice administered enterohemorrhagic *Escherichia coli* O157:H7 (EHEC) strain FRIK-2001 alone, histogram bars indicated with letters differ (*p* ≤ 0.018). Histogram bars indicated with asterisks differ (*p* ≤ 0.001). (**B**) Keratinocyte-derived cytokine (*Kc*). Histogram bars indicated with asterisks differ (*p* ≤ 0.001). (**C**) Transforming growth factor-beta (*Tgfb*). As there was no impact of the EHEC or commensal *E. coli* treatments, data were averaged over these factors. Histogram bars indicated with asterisks differ (*p* < 0.001).

## Data Availability

The WGS information for the 18 commensal *E. coli* strains were submitted to the Microbial Genomes database of NCBI under BioProject accession PRJNA935769.

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
