# Peer review of "Commensal Escherichia coli Strains of Bovine Origin Competitively Mitigated Escherichia coli O157:H7 in a Gnotobiotic Murine Intestinal Colonization Model with or without Physiological Stress"

_animals, 2023, doi:10.3390/ani13162577_

Round 1

Reviewer 1 Report

In this manuscript the authors aimed at developing a mice model mimicking cattle intestinal colonization by EHEC, and at examining the ability of a consortium of 18 bovine commensal E coli strains to outcompete EHEC O157:H7. Availability of models simulating the bacterial interactions encountered in cattle is important to develop and evaluate mitigation measures that could reduce carriage and shedding of EHEC by their main reservoir, i.e. cattle. Here the authors show that the administration of 18 commensal E. coli strains at the same time as EHEC O157:H7 effectively reduced intestinal densities of the pathogen in mice and improved its pathologic impacts on the host.

Major comments

The authors use the terms  “germ-free (GF) mice” (e.g. for inoculation of EHEC, L102-103)  and “gnotobiotic (GB) mice” (e.g. for inoculation of commensal E. coli strains, L105) as these terms were interchangeable throughout the manuscript. This is rather confusing. The authors should clarify this and provide definitions for GF and BG mice.

From the whole genome sequences obtained for the 18 E. coli commensal strains, the authors should provide additional useful genetic characteristics such as the phylogenetic groups, sequence types and serotypes, which are very informative (even more than SNP analysis). Such characterization can easily be performed through the use of online softwares (such as those provided by Enterobase or CGE). More importantly, the analysis of the whole genome sequences for bacteriocin genes should be performed as these compounds may have contributed strongly to EHEC reduction in mice. 

In control mice samples, the total E. coli density reached ca 9 log copy/g (as observed from the first two histograms in Fig S3).  It is unclear what was the origin of E. coli detected in those samples, as no EHEC nor commensal E. coli were administered to the corresponding mice. Could this E. coli population outcompete EHEC O157:H7 as well ?

Other comments

L60. The terms “enterohemorraghic Escherichia coli O157:H7 (EHEC)” could be used from the beginning of the introduction (and “EHEC O157:H7” afterwards).

L155. Were the five EHEC genomes sequenced in the frame of this study ? If yes, please provide accession numbers and describe the procedure for whole genome sequencing (as for commensal strains L183-186). If not, please provide a reference.

L231 and L248. Please refer to section 2.10 for the administration mode.

L267-282. Please change typography (no italic).

L275-276 (and L558-560). When examining the relative abundance of commensal strains, why using the labor intensive method PFGE for isolate characterization instead of high throughput PCR or qPCR ?

L283-290. Please indicate how many mice were used per cage.  

L364-375. Did the qPCR assay for quantification of total E. coli detect each of the 18 commensal E. coli strains ?

L535-537. Total absence of virulence genes and antibiotic resistance genes is quite surprising since at least a few genes are generally detected from E. coli strains, even commensal ones (such as genes encoding fimbrial adhesins, microcins/colicins, gad, etc… for virulence; and mdfA for antibiotic resistance). 

L559. Please add a table containing the list of strains with their frequency.

L603-605. “Tnfa and Kc mRNA was reduced in mice administered commensal E. coli” : compared to which condition ?  To EHEC administered mice ? 

L711. “agnotobiotic” =>  “gnotobiotic” ?

L795-797. Here, do the authors suggest that commensal E. coli could contribute to Shigatoxin production and severity of infection ? This should be stated more clearly. References could be added too (such as Iversen et al. 2015, doi: 10.3389/fcimb.2015.00005)

L847-849. Is the access to nutrients the sole hypothesis for the reduction of EHEC by commensal strains. Could there be alternative possibilities such as inhibitory compounds ? Did the whole genome sequences of the 18 commensal E. coli contain bacteriocin-encoding genes ?

L911-913. Is PRJNA935769 the correct accession number, as no items were found from NCBI using this term.

Figure 1 legend. How many mice were used per strain ?

Figure 2 legend. How many mucosa samples were analyzed per strain ?

Figure 3 legend. “lower histopathologic change score was observed in mice inoculated with FRIK-2001 at the day 8 endpoint”. Was this observation made in comparison to day 5 post-inoculation ? or to control mice (as indicated L472-773) ? Please clarify. 

Figure 7. Y axis: « EHEC density » would be more appropriate.

Author Response

Reviewer 1 Comments and Suggestions for Authors

Reviewer comment: In this manuscript the authors aimed at developing a mice model mimicking cattle intestinal colonization by EHEC, and at examining the ability of a consortium of 18 bovine commensal E coli strains to outcompete EHEC O157:H7. Availability of models simulating the bacterial interactions encountered in cattle is important to develop and evaluate mitigation measures that could reduce carriage and shedding of EHEC by their main reservoir, i.e. cattle. Here the authors show that the administration of 18 commensal E. coli strains at the same time as EHEC O157:H7 effectively reduced intestinal densities of the pathogen in mice and improved its pathologic impacts on the host.

Author response: No response required.

Major Comments

Reviewer comment: The authors use the terms “germ-free (GF) mice” (e.g. for inoculation of EHEC, L102-103) and “gnotobiotic (GB) mice” (e.g. for inoculation of commensal E. coli strains, L105) as these terms were interchangeable throughout the manuscript. This is rather confusing. The authors should clarify this and provide definitions for GF and BG mice.

Author response: Definitions for both germ-free (GF) and gnotobiotic (GB) have been added to the Introduction section, and we now limit the use of GF in favor of GB wherever possible (i.e., in the Results and Discussion sections).

Reviewer comment: From the whole genome sequences obtained for the 18 E. coli commensal strains, the authors should provide additional useful genetic characteristics such as the phylogenetic groups, sequence types and serotypes, which are very informative (even more than SNP analysis). Such characterization can easily be performed through the use of online softwares (such as those provided by Enterobase or CGE). More importantly, the analysis of the whole genome sequences for bacteriocin genes should be performed as these compounds may have contributed strongly to EHEC reduction in mice. 

Author response: We have completed additional analyses and we have added three supplemental Tables containing information on the predicted phenotypic characteristics of the 18 commensal E. coli strains (i.e., Table S4-S6). In addition, we have present a summary of the results of these analyses in the Results and Discussion sections.

Reviewer comment: In control mice samples, the total E. coli density reached ca 9 log copy/g (as observed from the first two histograms in Fig S3).  It is unclear what was the origin of E. coli detected in those samples, as no EHEC nor commensal E. coli were administered to the corresponding mice. Could this E. coli population outcompete EHEC O157:H7 as well?

Author response: We assume that the reviewer is referring to Figure S4. There are no control treatment mice samples portrayed in the graph (control treatment mice were germ free); all eight histogram bars represent total E. coli densities from mice that were inoculated with commensal E. coli. We assume that the confusion is with the first two histogram bars of the graph (EHEC ‒; Gown Communally – and Corticosterone ‒). These mice were inoculated with commensal E. coli that were grown individually. To prevent confusion, we have included “Grown separately” with corresponding “‒” or “+” signs.

Specific Comments

Reviewer comment: L60. The terms “enterohemorraghic Escherichia coli O157:H7 (EHEC)” could be used from the beginning of the introduction (and “EHEC O157:H7” afterwards).

Author response: Terms have been altered as suggested.

Reviewer comment: L155. Were the five EHEC genomes sequenced in the frame of this study? If yes, please provide accession numbers and describe the procedure for whole genome sequencing (as for commensal strains L183-186). If not, please provide a reference.

Author response: A citation referencing the sequences for the EHEC genomes used in the study has been added (Strachan et al. 2015).

Reviewer comment: L231 and L248. Please refer to section 2.10 for the administration mode.

Author response: Statements have been added as suggested.

Reviewer comment: L267-282. Please change typography (no italic).

Author response: The italicization of the text has been corrected.

Reviewer comment:  L275-276 (and L558-560). When examining the relative abundance of commensal strains, why using the labor intensive method PFGE for isolate characterization instead of high throughput PCR or qPCR?

Author response: We agree with the reviewer. The pulsed-field gel electrophoresis (PFGE) analysis of commensal strain frequency was completed before the whole genome sequence data was available, and thus before the strain-specific primers were developed. It was our intention to conduct quantitative PCR of the culture samples, but unfortunately the samples were inadvertently lost. None-the-less, the PFGE frequency data shows that all of the commensal E. coli isolates that were grown communally were present in the inoculum, and subsequent analysis of commensal E. coli strain abundance by qPCR in mice supports this observation.

Reviewer comment: L283-290. Please indicate how many mice were used per cage.  

Author response: This is now specified in the manuscript (i.e., “… in the IVC containing an individually housed mouse”).

Reviewer comment: L364-375. Did the qPCR assay for quantification of total E. coli detect each of the 18 commensal E. coli strains?

Author response: Yes, the primers utilized (HDA1/HDA2) target total bacteria. It is a universal primer set that targets conserved regions of DNA in the 16S rRNA gene. The GF mice were only inoculated with commensal E. coli bacteria or EHEC or both, any bacterium present will be detected by these primers. We have clarified this within the revised manuscript.

Reviewer comment: L535-537. Total absence of virulence genes and antibiotic resistance genes is quite surprising since at least a few genes are generally detected from E. coli strains, even commensal ones (such as genes encoding fimbrial adhesins, microcins/colicins, gad, etc… for virulence; and mdfA for antibiotic resistance). 

Author response: We have reanalyzed the genomes of the 18 commensal E. coli strains using a variety of bioinformatics tools as suggested, and we have added three supplemental tables, along with ancillary information in the Materials and Methods, Results, and Discussion sections. See our response to major comment #2 above.

Reviewer comment: L559. Please add a table containing the list of strains with their frequency.

Author response: A supplemental table with this information has been added as suggested (Table S7).

Reviewer comment: L603-605. “Tnfa and Kc mRNA was reduced in mice administered commensal E. coli”: compared to which condition?  To EHEC administered mice? 

Author response: Yes. This has been clarified in the revised manuscript.

Reviewer comment: L711. “agnotobiotic” => “gnotobiotic”?

Author response: “Agnotobiotic” has been replaced with “normal flora”.

Reviewer comment: L795-797. Here, do the authors suggest that commensal E. coli could contribute to Shigatoxin production and severity of infection? This should be stated more clearly. References could be added too (such as Iversen et al. 2015, doi: 10.3389/fcimb.2015.00005)

Author response: It is suggested that commensal E. coli can stimulate the production of shigatoxin from EHEC strains. Using a gnotobiotic murine colonization allows this to be ascertained, and we proposed that commensal strains can be evaluated in vivo using this colonization model before progressing to evaluations in cattle. A statement and the Inversen et al. 2015 has been added to the revised manuscript as suggested.

Reviewer comment: L847-849. Is the access to nutrients the sole hypothesis for the reduction of EHEC by commensal strains. Could there be alternative possibilities such as inhibitory compounds? Did the whole genome sequences of the 18 commensal E. coli contain bacteriocin-encoding genes?

Author response: Additional information on the possible role of antibiosis has been added to the revised manuscript.

Reviewer comment: L911-913. Is PRJNA935769 the correct accession number, as no items were found from NCBI using this term.

Author response: When we submitted the whole genome sequence data for the commensal E. coli strains, we embargoed it until March 31, 2024, or until publication of the paper, whichever came first. We have contacted NCBI and requested that the embargo be removed. NCBI indicated that it will be released July 29, 2023. Thus, the data should be available to the reviewer if they wish to examine it.

Reviewer comment: Figure 1 legend. How many mice were used per strain?

Author response: This information has been added to the figure legend (i.e., “n = three replicate mice per treatment per time post-inoculation”).

Reviewer comment: Figure 2 legend. How many mucosa samples were analyzed per strain?

Author response: One mucosa sample was analyzed per sample location per mouse. There were three replicate mice analyzed per treatment. This information has been added to the figure legend.

Reviewer comment: Figure 3 legend. “lower histopathologic change score was observed in mice inoculated with FRIK-2001 at the day 8 endpoint”. Was this observation made in comparison to day 5 post-inoculation? or to control mice (as indicated L472-773)? Please clarify. 

Author response: This has been clarified in the figure legend (i.e., “… relative to control treatment mice).

Reviewer comment: Figure 7. Y axis: « EHEC density » would be more appropriate.

Author response: The y-axis label has been changed as suggested.

Author general comment: We have added the following statement to the Acknowledgements section, “We also thank the two peer reviewers for their constructive comments”.

Reviewer 2 Report

The authors developed a GB mouse model to study the host-EHEC interaction in a manner that mimics the interaction in cattle. Is provides more animal models for studying EHEC infection in humans. However, the following issues and questions need to be solved.

>>>The author should consider adding a part for experimental design including the challenge/assessment/euthanasia at each time point in the M&M section

>>> The author should also use CFU for the quantification of E. coli from the mice colon, or should explain why only use qPCR.

>>> Figure 1 and its figure legend does not include detailed information on Health assessment scores, so it is every hard for readers to follow and get the information directly from the figure.

>>>The representative pictures for H&E in figure 3 are missing.

>>>Why only study include Metabolite Profiles in the kidney and liver, but not the intestinal tissues?

>>>Line 267-282, the font need to be consistent.

>>>Line 335, “the” should be added before cecum

Minor editing of English language required

Author Response

Reviewer 2 Comments and Suggestions for Authors

Reviewer comment: The authors developed a GB mouse model to study the host-EHEC interaction in a manner that mimics the interaction in cattle. Is provides more animal models for studying EHEC infection in humans. However, the following issues and questions need to be solved.

Author response: See responses below.

Reviewer comment: The author should consider adding a part for experimental design including the challenge/assessment/euthanasia at each time point in the M&M section

Author response: We are not entirely sure what the reviewer is requesting, but think that they are suggesting consolidation of challenge/assessment/euthanasia into a common subheading? As two major experiments were conducted with experimental characteristic unique to each (i.e., EHEC strain selection, and competitive exclusion by commensal E. coli strains), coupled with current length of the Materials and Methods section (i.e., 6.5 pages), we feel that the information on challenge/assessment/euthanasia is best presented in separate subsections. To aid in the understanding of the experimental design for the competitive exclusion experiment (i.e., experiment 2), which is a factorial experiment, we have included a supplemental figure to pictorially illustrate the experimental design and timelines (i.e., Figure S1).

Reviewer comment: The author should also use CFU for the quantification of E. coli from the mice colon, or should explain why only use qPCR.

Author response: We used qPCR to quantify E. coli because it was the most specific and sensitive method, we had available to quantify and differentiate different E. coli strains within the same sample (EHEC and commensal E. coli). This was not possible using culturomic methods given that selective isolation of individual E. coli strain from the same sample is not possible. For experiment two, a single EHEC (FRIK 2001) and 18 commensal E. coli strains had to be differentially identified and quantified within the same sample. Utilizing strain-specific primers allowed us to quantify each strain individually. Moreover, we decided to utilize qPCR to quantify E. coli in all of the experimental steps of the project in order to keep consistency of quantification (with the exception of the quantification of commensal E. coli strains within the culture medium when grown communally). We initially characterized commensal E. coli frequency within the culture medium using isolation and pulsed-field gel electrophoresis (PFGE). This strategy is relatively labor-intensive and less accurate than qPCR. The PFGE analysis of commensal strain frequency was completed before the whole genome sequence data was available, and thus before the strain-specific primers were developed. It was our intention to conduct quantitative PCR of the culture samples, but unfortunately the samples were inadvertently lost. None-the-less, the PFGE frequency data shows that all of the commensal E. coli isolates that were grown communally were present in the inoculum, and subsequent analysis of commensal E. coli strain abundance by qPCR in mice confirms this.

Reviewer comment: Figure 1 and its figure legend does not include detailed information on Health assessment scores, so it is every hard for readers to follow and get the information directly from the figure.

Author response: We have added details on the health assessment scores to the figure legend as suggested.

Reviewer comment: The representative pictures for H&E in figure 3 are missing.

Author response: The histologic changes that we observed were typical of those described in previous publications of EHEC O157:H7 in mice. Given that the manuscript is quite extensive with nine figures, six supplemental figures, and seven tables, we feel that inclusion of representative micrographs probably not warranted. The pathologist who conducted the histopathologic change scoring is a board-certified pathologist who was blinded to treatment. This being said if the reviewer feels strongly that we should include representative micrographs we can do so.

Reviewer comment: Why only study include Metabolite Profiles in the kidney and liver, but not the intestinal tissues?

Author response: The overarching goal of utilizing metabolomics was to observe the systemic impact of EHEC O157:H7 on the host. The liver is one of the main organs to present metabolomic changes in mice given that nutrient and toxin absorption from the intestinal lumen into the blood system will reach the liver first. Likewise, the impact of EHEC O157:H7 on kidneys is well documented due to Shiga toxin toxicity. We were not certain that either organ was going to be affected by changes that were large enough to be observed by histopathological analysis. Given the ability of metabolomics to detect minute impacts of a pathogen on liver and kidney, applying metabolomics was the option we considered. In contrast to impacts on the kidney and liver, impacts of infection by EHEC O157:H7 in intestinal tissues are comparatively more pronounced, and can be measured by histopathological analysis. We did consider examining the metabolome of intestinal tissues. However, given that limited amount of colonic tissue available (i.e., the animal model is very small), we felt that there was inadequate tissue to do metabolomics, and we decided to prioritize histopathologic and inflammation marker analysis in the current study. Metabolomic analysis of intestinal samples could be conducted in subsequent experimentation.

Reviewer comment: Line 267-282, the font need to be consistent.

Author response: The italicization of the text has been corrected.

Reviewer comment: Line 335, “the” should be added before cecum

Author response: “The” has been added.

Author general comment: We have added the following statement to the Acknowledgements section, “We also thank the two peer reviewers for their constructive comments”.

Round 2

Reviewer 1 Report

I thank the authors for addressing all my comments and for improving their manuscript.

Further to the added results obtained from whole genome analysis of commensal strains, one point (below) would deserve particular attention. 

L563-565. This strain is the only one belonging to phylogroup B2 and possesses genomic characteristics similar to those of the probiotic E. coli Nissle strain, such as B2 phylotype, carriage of iron acquisition systems yersiniabactin (fyuA) and salmochelin (iroN), colibactin genotoxin (clb / pks island) and microcin M/H47 encoding genes. These characteristics are worth mentioning, here and most importantly in the discussion L834-835, especially as E. coli Nissle was demonstrated to reduce animal colonization by pathogens, including EHEC (e.g. Maltby R, et al. PloS one. 2013;8(1):e53957).

Minor comments.

L552. different animals.

L555-557. B1 was the main phylogroup (11 out of 18 strains), this point is worth mentioning.

L557. identified carriage (please delete "that")

L946. Here, table S4 should probably be table S7. Please change and add also tables S4-S6 about predicted phenomics.

Author Response

Comments and Suggestions for Authors

Reviewer comment: I thank the authors for addressing all my comments and for improving their manuscript.

Author response: We sincerely appreciate the valuable comments/suggestions provided by the reviewer.

Major Comment

Reviewer comment: Further to the added results obtained from whole genome analysis of commensal strains, one point (below) would deserve particular attention.

L563-565. This strain is the only one belonging to phylogroup B2 and possesses genomic characteristics similar to those of the probiotic E. coli Nissle strain, such as B2 phylotype, carriage of iron acquisition systems yersiniabactin (fyuA) and salmochelin (iroN), colibactin genotoxin (clb / pks island) and microcin M/H47 encoding genes. These characteristics are worth mentioning, here and most importantly in the discussion L834-835, especially as E. coli Nissle was demonstrated to reduce animal colonization by pathogens, including EHEC (e.g. Maltby R, et al. PloS one. 2013;8(1):e53957).

Author response: Thank you for the suggestions. The information has been added to the manuscript.

Minor Comments.

Reviewer comment: L552. different animals.

Author response: The typo has been corrected.

Reviewer comment: L555-557. B1 was the main phylogroup (11 out of 18 strains), this point is worth mentioning.

Author response: A statement has been added to the Results section.

Reviewer comment: L557. identified carriage (please delete "that")

Author response: “that” has been deleted.

Reviewer comment: L946. Here, table S4 should probably be table S7. Please change and add also tables S4-S6 about predicted phenomics.

Author response: We confirm that all of the supplemental tables are cited appropriately and in the correct order. 

Reviewer 2 Report

No more comments

Author Response

Comments and Suggestions for Authors

Reviewer comment: No more comments

Author response: We thank the reviewer for providing previous comments/suggestions that improved the manuscript.